# Online Decision Mediation

**Daniel Jarrett**[1]  **Alihan Hüyük**[1]  **Mihaela van der Schaar**[1,2,3]

Department of Applied Mathematics and Theoretical Physics
[1]University of Cambridge, [2]UCLA, [3]Alan Turing Institute

[dkj25,ah2075,mv472]@cam.ac.uk

## Abstract

Consider learning a decision support assistant to serve as an intermediary between (oracle) expert behavior and (imperfect) human behavior: At each time, the algorithm observes an action chosen by a fallible agent, and decides whether to *accept* that agent's decision, *intervene* with an alternative, or *request* the expert's opinion. For instance, in clinical diagnosis, fully-autonomous machine behavior is often beyond ethical affordances, thus real-world decision support is often limited to monitoring and forecasting. Instead, such an intermediary would strike a prudent balance between the former (purely prescriptive) and latter (purely descriptive) approaches, while providing an efficient interface between human mistakes and expert feedback. In this work, we first formalize the sequential problem of *online decision mediation* —that is, of simultaneously learning and evaluating mediator policies from scratch with *abstentive feedback*: In each round, deferring to the oracle obviates the risk of error, but incurs an upfront penalty, and reveals the otherwise hidden expert action as a new training data point. Second, we motivate and propose a solution that seeks to trade off (immediate) loss terms against (future) improvements in generalization error; in doing so, we identify why conventional bandit algorithms may fail. Finally, through experiments and sensitivities on a variety of datasets, we illustrate consistent gains over applicable benchmarks on performance measures with respect to the mediator policy, the learned model, and the decision-making system as a whole.

## 1 Introduction

Research in data-driven decision support has burgeoned in recent years, with proposed applications in a wide variety of domains such as finance [1], psychology [2], and medicine [3]. Most work on machine learning for decision support falls into two categories: On one hand, *descriptive* approaches deal with monitoring, forecasting, and learning interpretable parameterizations of observed human behavior [4–8]. While such tools can help audit and debug decision-making, they play a limited role in directly guiding human behavior. On the other hand, *prescriptive* approaches deal with systems that behave autonomously, optimally, and with minimal manual control [9–12]. While such tools can reduce the need for expert input, they are often at odds with ethical considerations—especially in high-stakes settings such as healthcare [13–18]. Instead, we argue for a third: We believe machine learning decision support has a viable role as an *intermediary* between (oracle) expert behavior and (imperfect) human behavior—that is, as an "assistant", which would strike a prudent balance between the above two approaches, while providing an efficient interface between human mistakes and expert feedback.

**Online Decision Mediation**  In this paper, we consider learning and evaluating *mediator policies* for online decision support from scratch: At each time step, upon observing the context vector of an incoming instance, the mediator policy decides whether to *accept* the human's action, *intervene* with its own output, or *request* the expert's opinion—and this determines which action is ultimately taken. Deferring to the oracle obviates the risk of error, but incurs an upfront penalty, and reveals the otherwise hidden expert action as a new training data point. As our running example, consider the task of

36th Conference on Neural Information Processing Systems (NeurIPS 2022).

early diagnosis in Alzheimer's disease, where the action is to diagnose each incoming patient as cognitively normal, mildly impaired, or at risk of dementia [19,20]. Our problem setting is distinguished primarily by three key characteristics: (1) Instances—i.e. patients—arrive in a *streaming* process, and actions must be taken immediately and sequentially. (2) Feedback—i.e. true diagnoses—is only available in an *abstentive* manner, meaning ground truths are only revealed if the oracle is deferred to. (3) Evaluation—i.e. cumulative regret—is computed in an *online* fashion, without convenient separation into "training" versus "testing" phases. This setting is challenging but general, and applicable wherever domain experts are resource-constrained, e.g. if the costs of definitive examinations are high.

**Contributions** In the sequel, we first formalize this sequential problem of *online decision mediation* ("ODM"), and establish its unique challenges versus more conventional problem settings (Section 2). Second, we identify why conventional bandit algorithms may fail, and describe our proposed solution, *uncertainty-modulated policy for intervention and requisition* ("UMPIRE"), which seeks to trade off (immediate) loss terms against expected (future) improvements in generalization error (Section 3). Finally, through experiments and sensitivities on a variety of real-world datasets, we illustrate consistent gains over applicable benchmarks on a comprehensive set of performance measures with respect to the mediator policy, the learned model, and the entire decision-making system as a unit (Section 4).

**Implications** Humans are heterogeneous, and mistakes require timely correction. Machines are also fallible, and models require timely learning. The implications are clear: Rather than pitting computers *against* clinicians, an efficient mediator should *augment* clinician capabilities by leveraging costly but informative expert resources. By focusing on the (human-expert-mediator) decision-making system as a whole, we take a first step towards more methodical integration of machines "into the loop". Moreover, the technical problem itself combines diverse challenges from sequential decision-making, learning with rejection, and active learning, thus opening the door to multiple avenues of further work.

## 2 Online Decision Mediation

### 2.1 Problem Formulation

We use uppercase for random variables, and lowercase for specific values. Let $X$ denote the input variable, taking on values $x \in \mathcal{X}$, and let $Y$ denote the target variable, taking on values $y \in \mathcal{Y}$. In line with related fields, $X$ may synonymously be referred to as "contexts", "features", and "states" depending on the underlying task, and $Y$ may likewise be referred to as "actions", "decisions", and "labels". Per our motivating example, we shall adopt the "context-action" terminology for consistency, although our framework applies to any decision task that requires mapping inputs to specific outputs. Following most related settings, we focus on discrete action spaces, and leave continuous actions for future work.

**Human, Expert, and Mediator** Let $\rho(X)$ denote an exogenous distribution from which a streaming sequence of contexts $\{X_t\}_t$ is drawn and indexed by time step. We consider three decision-makers: a human, an expert, and a mediator. In each round, a human action is drawn as $\tilde{Y}_t \sim \tilde{\pi}(\cdot|X_t)$ from an (unknown and possibly stochastic) *human policy* $\tilde{\pi} \in \Pi$. For instance, this is the noisy diagnosis issued by an apprentice clinician. If prompted, an expert action may likewise be drawn as $Y_t \sim \pi_*(\cdot|X_t)$ from an (unknown and possibly stochastic) *expert policy* $\pi_* \in \Pi$. For instance, this is the final diagnosis issued by a senior doctor—that is, should they indeed be appointed to conduct a full examination of the patient. Finally, a "mediator" is identified by the tuple $(\hat{\pi}, \phi)$, consisting of a (learned) *model policy* $\hat{\pi} \in \Pi$—from which model actions $\hat{Y}_t \sim \hat{\pi}(\cdot|X_t)$ may be drawn—as well as a *mediator policy* $\phi \in \Phi$:

**Definition 1 (Decision System)** Let $\mathcal{S} := (\tilde{\pi}, \hat{\pi}, \pi_*, \phi)$ denote the *decision system* as a whole. Given an incoming $(X, \tilde{Y})$, the mediator policy defines a distribution $\phi(\cdot|X, \tilde{Y})$ over the space of mediator actions $\mathcal{Z} := \{0, 1, 2\}$, consisting of options *accept* ($z=0$), *intervene* ($z=1$), and *request* ($z=2$). Let $\delta(Y-y)$ be the Dirac delta centered at $y$; drawing $Z \sim \phi(\cdot|X, \tilde{Y})$ induces the overall *system policy*:

$$\pi_{\mathcal{S}}(\cdot|X, \tilde{Y}) := \mathbb{1}_{[Z=0]}\delta(Y - \tilde{Y}) + \mathbb{1}_{[Z=1]}\delta(Y - \arg\max_y \hat{\pi}(y|X)) + \mathbb{1}_{[Z=2]}\pi_*(\cdot|X) \qquad (1)$$

Intervening incurs some cost $k_{\text{int}}$ (e.g. inconveniencing the apprentice clinician to reconsider/alter their decision). Requesting incurs some cost $k_{\text{req}}$ (e.g. appointing the senior doctor to provide their opinion), but also reveals the otherwise hidden ground-truth action: Let $D_t := \{(X_\tau, Y_\tau) : Z_\tau = 2\}_{\tau=1}^t$ denote the cumulative dataset of requested points, taking on values $d_t \in \mathcal{D}_t := \cup_t(\mathcal{X} \times \mathcal{Y})^t$, and constitutes the training set with which the model policy $\hat{\pi}$ is defined. Thus feedback (for learning) is "abstentive" in that it is only observable when the system "abstains" in favor of deferring the decision to the expert.

**Risk and Evaluation**  Among other aspects, our objective of interest differs from supervised learning in two important ways: First, we are chiefly interested in the performance of the decision system $\mathcal{S}$, instead of the model $\hat{\pi}$ per se. Second, learning and evaluation are *both* conducted online. By way of contrast, consider first the familiar supervised learning objective, which is simply concerned with minimizing the *generalization error* of the model over the underlying data distribution, or the "model risk":

$$\mathcal{R}(\hat{\pi}) \coloneqq \mathbb{E}_{\substack{X \sim \rho \\ Y \sim \pi_*(\cdot|X)}} \ell(Y, \hat{\pi}(\cdot|X)) \tag{2}$$

where $\ell : \mathcal{Y} \times \Delta(\mathcal{Y}) \to \mathbb{R}$ is some choice of loss function. In decision problems, this most commonly takes the form of the zero-one loss $\ell(Y, \hat{\pi}(\cdot|x)) \coloneqq \ell_{01}(Y, \arg\max_y \hat{\pi}(y|x))$, or in some cases—if a surrogate loss is required—the cross-entropy $\ell(Y, \hat{\pi}(\cdot|x)) \coloneqq \log \hat{\pi}(Y|x)$; we shall use the former to be consistent with comparable literature. Now, our main focus is not the model risk, but the "system risk":

**Definition 2 (System Risk)**  Let $\mathcal{M} \coloneqq (\mathcal{X}, \mathcal{Y}, \tilde{\pi}, \pi_*, \rho, k_{\text{int}}, k_{\text{req}})$ denote the *mediation setting*. Given a mediator $(\hat{\pi}, \phi)$, the *system risk* in each round $t$ is the expected error of the induced system policy (i.e. having selected from human, model, and expert actions), plus the upfront cost of mediator actions:

$$\mathcal{R}_t(\hat{\pi}, \phi) \coloneqq \mathbb{E}_{\substack{X_t \sim \rho \\ \tilde{Y}_t \sim \tilde{\pi}(\cdot|X_t) \\ Y_t \sim \pi_*(\cdot|X_t)}} \big[ \phi(Z_t = 0 | X_t, \tilde{Y}_t) \ell(Y_t, \delta(Y - \tilde{Y}_t)) + \\ \phi(Z_t = 1 | X_t, \tilde{Y}_t)(\ell(Y_t, \hat{\pi}(\cdot|X_t)) + k_{\text{int}}) + \phi(Z_t = 2 | X_t, \tilde{Y}_t) k_{\text{req}} \big] \tag{3}$$

Then the *online decision mediation* ("ODM") problem is to select a mediator $(\hat{\pi}, \phi)$ to minimize (cumulative) *regret* over a possibly-unspecified horizon. Importantly, note that this is a more challenging objective than simply minimizing the generalization error of the model, system, or some asymptotic complexity thereof: Here we have no separation between "training" versus "testing", since losses begin accumulating from the very first step of the sequential process. The regret at any round $n$ is given by:

$$\mathbf{Regret}(\hat{\boldsymbol{\pi}}, \boldsymbol{\phi})[n] \coloneqq \sum_{t=0}^{n} \big( \mathcal{R}_t(\hat{\pi}_t, \phi_t) - \mathcal{R}_t(\pi_*, \phi_*) \big) \tag{4}$$

where we assume realizability such that the best-in-class mediator is defined as the tuple consisting of the expert policy $\pi_*$ and the greedy mediator policy $\phi_*$ (i.e. always choosing $Z$ to minimize each round's immediate system risk). Note the above notation makes it explicit that both the model policy and mediator policy evolve as sequences $\hat{\boldsymbol{\pi}} \coloneqq \{\hat{\pi}_t\}_t$ and $\boldsymbol{\phi} \coloneqq \{\phi_t\}_t$ that—in general—depend on $D_t$.

**Remark 1 (Assumptions)**  For ease of exposition, we assume all mistakes are equally important, that $k_{\text{int}}, k_{\text{req}}$ are constants, and expert action classes are more or less balanced over the input distribution; allowing relaxations is straightforward and left for future work. To eliminate the more trivial cases, we assume $0 < k_{\text{int}} < k_{\text{req}}$, and operate in the common rejection regime where $k_{\text{req}} = \frac{m}{m-1} - \gamma$ for some small $\gamma > 0$, with $m$ being the number of actions in $\mathcal{Y}$; this induces the most interesting tradeoff setting where abstention is neither excessive nor immediately ruled out by the greedy policy. (In our experiments, we shall empirically consider a range of sensitivities). We assume nothing about $\tilde{\pi}$, for instance if it is even stationary. Lastly, we assume $D_0$ is randomly seeded with one example per action class.

## 2.2  Related Work

The ODM problem lies at the confluence of three classes of learning problems while being distinct from all: (i) learning with rejection, (ii) stream-based active learning, and (iii) stochastic contextual bandits. As before, we employ generic notation and make note of synonymous terminology as appropriate.

**Learning with Rejection**  Compared to standard supervised learning, *learning with rejection* is a problem setting that endows the algorithm—during test time—with the option to "reject" their own prediction in favor of expert advice [21–26]. This is variously referred to as the option to "abstain" from a decision, or to "defer" to an oracle. Exercising it incurs an *abstention cost* $k_{\text{abs}}$, but enables avoiding misclassification when the model is uncertain. Typically, a solution consists of a model policy $\hat{\pi}$ and a *rejection policy* $\psi$ defining a distribution $\psi(U|X)$ over the space of actions $\mathcal{U} \coloneqq \{1, 2\}$, consisting of the options *not abstain* ($u = 1$) and *abstain* ($u = 2$). While the rejection option is similar to that in ODM, learning proceeds from a static dataset, evaluation focuses on model risk, and—like supervised learning—labels in the batch are always available. An *online* variant of this setting shares more similarity with the ODM problem by focusing on minimizing the cumulative loss over the course of learning, instead of simply minimizing the held-out performance of the algorithm [27–30]. However, a key distinction from us is that feedback is *not* an active choice, and expert labels are always streamed (or when the model does *not* defer to the expert, which is exactly the opposite of our setting).

Table 1: *Online Decision Mediation vs. Related Work*. The ODM problem is distinguished by three key factors: (1) Learning is stream-based (so exploratory considerations cannot benefit from any pool-based comparison). (2) Feedback is abstentive (so some exploitative actions—precisely, $z \in \{0, 1\}$—yield no learning signal at all). (3) Evaluation is online (so the exploration-exploitation tradeoff is explicitly measured by the cumulative loss). Subscripts $t$ are omitted from policy terms. Shaded terms denote those evaluated by the risk function, "a.s.c." denotes asymptotic sample complexity, "feedback condition" indicates when ground-truths are revealed for learning, and "multi-class" indicates whether each setting is not restricted (theoretically or empirically) to binary decisions.

| Problem Setting | Stream-based | Abstain Option | Active Request | Components (Evaluated) | Risk Function of Interest | Minimization of Interest | Feedback Condition | Online Eval. | Multi-class |
|---|---|---|---|---|---|---|---|---|---|
| Supervised Learning | ✗ | ✗ | ✗ | $\pi_*, \hat{\pi}$ | $\mathcal{R} = \mathbb{E}_{\substack{X \sim \rho \\ Y \sim \pi_*(\cdot\mid X)}} \ell(Y, \hat{\pi}(\cdot\mid X))$ | $\mathcal{R}(\hat{\pi})$ | (n/a) | ✗ | ✓ |
| Learning with Rejection [21-26] | ✗ | ✓ | ✗ | $\pi_*, \hat{\pi}, \psi$ | $\mathcal{R} = \mathbb{E}_{\substack{X \sim \rho \\ Y \sim \pi_*(\cdot\mid X)}} [\psi(1\mid X)\ell(Y, \hat{\pi}(\cdot\mid X)) + \psi(2\mid X)k_{\text{abs}}]$ | $\mathcal{R}(\hat{\pi}, \psi)$ | (n/a) | ✗ | ✓ |
| Online Learning with Rejection [27-30] | ✓ | ✓ | ✗ | $\pi_*, \hat{\pi}, \psi$ | $\mathcal{R}_t = \mathbb{E}_{\substack{X_t \sim \rho \\ Y_t \sim \pi_*(\cdot\mid X_t)}} [\psi(1\mid X_t)\ell(Y_t, \hat{\pi}(\cdot\mid X_t)) + \psi(2\mid X_t)k_{\text{abs}}]$ | $\sum_t (\mathcal{R}_t(\hat{\pi}, \psi) - \mathcal{R}_t(\pi_*, \psi_*))$ | (always) | ✓ | ✗ |
| (Stream-based) Active Learning [31-36] | ✓ | ✗ | ✓ | $\pi_*, \hat{\pi}, \phi$ | $\mathcal{R} = \mathbb{E}_{\substack{X \sim \rho \\ Y \sim \pi_*(\cdot\mid X)}} \ell(Y, \hat{\pi}(\cdot\mid X))$ | $\mathcal{R}(\hat{\pi})$ a.s.c. | $Z = 2$ | ✗ | ✗ |
| Active Learning with Abstention [37-40] | ✓ | ✓ | ✓ | $\pi_*, \hat{\pi}, \psi, \phi$ | $\mathcal{R} = \mathbb{E}_{\substack{X \sim \rho \\ Y \sim \pi_*(\cdot\mid X)}} [\psi(1\mid X)\ell(Y, \hat{\pi}(\cdot\mid X)) + \psi(2\mid X)k_{\text{abs}}]$ | $\mathcal{R}(\hat{\pi}, \psi)$ a.s.c. | $Z = 2$ | ✗ | ✗ |
| Dual Purpose Learning [41] | ✓ | ✓ | ✓ | $\pi_*, \hat{\pi}, \phi = \psi$ | $\mathcal{R} = \mathbb{E}_{\substack{X \sim \rho \\ Y \sim \pi_*(\cdot\mid X)}} [\ell(Y, \hat{\pi}(\cdot\mid X)) \mid Z = 1]$ | $\mathcal{R}(\hat{\pi}, \phi)$ a.s.c. | $Z = 2$ | ✗ | ✗ |
| (Stochastic Contextual) Bandits [42-48] | ✓ | ✗ | ✗ | $\upsilon, \hat{\pi}$ | $\mathcal{R}_t = -\mathbb{E}_{\substack{X_t \sim \rho \\ \hat{Y}_t \sim \hat{\pi}(\cdot\mid X_t)}} \upsilon(X_t, \hat{Y}_t)$ | $\sum_t (\mathcal{R}_t(\hat{\pi}) - \mathcal{R}_t(\pi_*))$ | (always) | ✓ | ✓ |
| Bandits with Active Learning [49-52] | ✓ | ✗ | ✓ | $\upsilon, \hat{\pi}, \phi$ | $\mathcal{R}_t = \mathbb{E}_{\substack{X_t \sim \rho \\ \hat{Y}_t \sim \hat{\pi}(\cdot\mid X_t)}} [\phi(2\mid X_t)k_{\text{req}} - \upsilon(X_t, \hat{Y}_t)]$ | $\sum_t (\mathcal{R}_t(\hat{\pi}) - \mathcal{R}_t(\pi_*))$ | $Z = 2$ | ✓ | ✓ |
| Apple Tasting with Context [53-55] | ✓ | ✗ | ✓ | $\upsilon, \phi = \hat{\pi}$ | $\mathcal{R}_t = \mathbb{E}_{X_t \sim \rho} [\phi(2\mid X_t)(k_{\text{req}} - \upsilon(X_t))]$ | $\sum_t (\mathcal{R}_t(\phi) - \mathcal{R}_t(\phi_*))$ | $Z = 2$ | ✓ | ✗ |
| Reinforced Active Learning [56-58] | ✓ | ✗ | ✓ | $\pi_*, \hat{\pi}, \phi$ | $\mathcal{R}_t = \mathbb{E}_{\substack{X_t \sim \rho \\ Y_t \sim \hat{\pi}(\cdot\mid X_t)}} [\phi(2\mid X_t)\ell(Y, \hat{\pi}(\cdot\mid X))]$ | $\sum_t (\mathcal{R}_t(\phi) - \mathcal{R}_t(\phi_*))$ | (always) | ✓ | ✓ |
| **Online Decision Mediation** | ✓ | ✓ | ✓ | $\pi_*, \tilde{\pi}, \hat{\pi}, \phi = \psi$ | $\mathcal{R}_t = \mathbb{E}_{\substack{X_t \sim \rho \\ \tilde{Y}_t \sim \tilde{\pi}(\cdot\mid X_t) \\ Y_t \sim \pi_*(\cdot\mid X_t)}} [\phi(0\mid X_t, \tilde{Y}_t)\ell(Y_t, \delta(Y - \tilde{Y}_t)) + \phi(1\mid X_t, \tilde{Y}_t)(\ell(Y_t, \hat{\pi}(\cdot\mid X_t)) + k_{\text{int}}) + \phi(2\mid X_t, \tilde{Y}_t)k_{\text{req}}]$ | $\sum_t (\mathcal{R}_t(\hat{\pi}, \phi) - \mathcal{R}_t(\pi_*, \phi_*))$ | $Z = 2$ | ✓ | ✓ |

**Stream-based Active Learning** In contrast with standard incremental learning, ground truths in *stream-based active learning* are unobserved unless actively "acquired" by the algorithm during training [31–36]. This is variously referred to as the option to "request" or "query" the oracle for its decision. Like supervised learning, the goal is to minimize test-time model risk, but with emphasis on reducing labeled and/or unlabeled asymptotic sample complexity. Typically, a solution consists of a model $\hat{\pi}$ and an *acquisition policy* $\phi$ defining a distribution $\phi(Z\mid X)$ over the space of actions $\mathcal{Z} := \{1, 2\}$, consisting of the options *not request* ($z = 1$) and *request* ($z = 2$). While the active request aspect is similar to that in ODM, evaluation focuses on model risk, so $\phi$ is not evaluated in the objective itself; moreover, the model has no ability to abstain from a prediction. One variant includes *abstention* to enable the algorithm—during test time—to reject their own prediction [37–40]. However, the objective remains to minimize test-time loss and its asymptotic complexity, which contrasts with our focus on evaluating cumulative losses over the entire process. A second variant called *dual purpose learning* is perhaps more similar in that the model abstains from a decision just when the expert is queried for its decision [41] (i.e. the acquisition policy $\phi$ coincides with the rejection policy $\psi$, and $\mathcal{Z} = \mathcal{U}$). But the risk function of interest is still like the usual test-time model risk, but now conditioned on $Z = 1$, so $\phi$ only enters the objective as a conditioning term to omit points where the model abstains.

**Stochastic Contextual Bandits** Lastly, ODM bears resemblance to stochastic contextual bandits, a class of sequential decision problems with the goal of minimizing (cumulative) *regret* defined in terms of an arbitrary "reward" function $\upsilon : \mathcal{X} \times \mathcal{Y} \to \mathbb{R}$ [42–48]. A main difference from ODM is that bandit rewards and are always observed as feedback for learning after each round, but no "expert" actions (viz. best-in-class policy) are available to be queried. One variant incorporates elements of *active learning* by stipulating that feedback must be requested at some cost $k_{\text{req}}$ [49–52]; in a special case dubbed *apple tasting*, model decisions are tied to acquisition decisions (i.e. the acquisition policy $\phi$ coincides with the model policy $\hat{\pi}$, and $\mathcal{Z} = \mathcal{Y}$) [53–55]. But unlike ODM, the model has no option to abstain, and there is no tradeoff between requesting information and making predictions. A second variant turns around and treats active learning itself as a bandit problem [56–58]; in the streaming, contextual case [56], the model policy $\hat{\pi}$ itself is actually not evaluated at all: Instead, the acquisition policy $\phi$ is rewarded positively just when querying the expert turns out to be "useful" (i.e. $Y \neq \hat{Y}$), and punished just when querying the expert turns out to be "redundant" (i.e. $Y = \hat{Y}$); moreover, feedback (for $\phi$, in this setting) is always observed. While these works are similar to ODM in focusing on the online evaluation objective of cumulative regret, the essential element of abstentive feedback—which is central to our motivation for a solution to ODM to mediate *efficiently* using expert resources—is missing.

Table 1 contextualizes ODM versus related work: Our setting combines the challenges from each, and is uniquely characterized by the three key aspects alluded to in Section 1: streaming instances, abstentive feedback, and online evaluation. In Section 3, we argue that a good algorithm must appropriately account for these challenges simultaneously. In Section 4, we verify empirically that neglecting any of them results in poor performance. (Additionally, since we are motivated from the perspective of decision support as an intermediary between humans, models, and experts, note that—as another practical component—ODM also extends $\mathcal{Z}$ with the option to accept human decisions $\tilde{Y} \sim \tilde{\pi}(\cdot|X)$.)

## 3 Mediator Policies

In light of the preceding discussion, it is clear a good mediator should satisfy the following criteria. The first deals with immediate loss, the second with future loss, and the third with trading off the two:

- It should accept or intervene only when *errors* are thereby unlikely (cf. learning with rejection).
- It should acquire ground truths when *uncertainty* may thereby be reduced (cf. active learning).
- It should balance such exploration and exploitation *adaptively* over time (cf. contextual bandits).

**Greedy Mediator** It is instructive to first examine the greedy policy. Given any model policy $\hat{\pi}$, the greedy mediator policy $\phi_*$ simply chooses $Z$ to minimize the immediate (i.e. one-step) system risk, which balances immediate probabilities of error with immediate costs of intervention/requisition:

$$
\begin{aligned}
\phi_*(Z|X,\tilde{Y}) := \delta\big(Z - \arg\min\nolimits_z \big(\mathbb{1}_{[z=0]}(1 - \hat{\pi}(\tilde{Y}|X)) \\
+ \mathbb{1}_{[z=1]}(1 - \hat{\pi}(\hat{Y}|X) + k_{\text{int}}) + \mathbb{1}_{[z=2]}k_{\text{req}}\big)\big)
\end{aligned}
\tag{5}
$$

where $\delta(Z-z)$ denotes the Dirac delta centered at $z$, and $\hat{Y} := \arg\max_y \hat{\pi}(y|X)$. It is clear that such a mediator policy is optimal in terms of regret if the model policy were already perfect (i.e. if $\hat{\pi} = \pi_*$), or if the model policy were otherwise fixed (e.g. if $Z=2$ no longer provided any feedback for learning).

**Passive Exploration** But what if the model is not perfect, and must incorporate new data points for learning? Actually, the greedy policy already "inadvertently" performs a sort of *passive* exploration: Whenever the target probabilities $[\hat{\pi}(y=1|X), ..., \hat{\pi}(y=m|X)]$ for a context $X$ are not sufficiently concentrated, $\phi_*$ would request from the expert, which may reduce uncertainty for points similar to $X$. However, $\phi_*$ may learn too slowly: If—at any point—the model is even *slightly* erroneously confident (e.g. $\hat{\pi}(y'|X) \geq 1 - k_{\text{req}} + \varepsilon$ for any $y' \neq y$, for any $\varepsilon > 0$), then it would simply not query the expert, and may commit similar mistakes again later. This is because it fails to distinguish between *aleatoric* and *epistemic* uncertainty: Not only do we wish to defer (viz. abstain) when the former is high at $X$, but we also wish to defer (viz. learn) when the latter may be reduced by knowing the ground truth at $X$. So the question is: Can we explore in a manner that better balances these (present vs. future) demands?

### 3.1 Bandit Mediator Policies

Prima facie, this resembles a bandit tradeoff, so the immediate question becomes: Can we simply formulate this as a specific instance of contextual bandits? Consider an ODM problem with $m$ actions in $\mathcal{Y}$: This gives $m + 2$ "arms" in total (i.e. an *accept*, a *request*, and an *intervene* arm for each of the $m$ underlying actions). However, precisely due to the nature of abstentive feedback, the answer is *no*:

**Loss vs. Feedback** In conventional bandit problems, there is no distinction between the notions "loss" and "feedback". In each round, when an arm $y_t$ is pulled in response to a context $x_t$, the negative reward $-\upsilon(x_t, y_t)$ serves dual purposes: It is always *incurred* into the performance measure (viz. regret), and it is always *observed* as a new data point for learning (viz. reinforcement). In ODM, however, "loss" and "feedback" are distinct quantities: In each round, when a mediator action $z_t$ is chosen in response to a context-action pair $(x_t, \tilde{y}_t)$, the resulting loss is always *incurred* (viz. Definition 2), but no information whatsoever (i.e. not even the loss) is *observed* as feedback for learning unless $z_t = 2$.

**Exploring without Learning** It should now be apparent why bandit algorithms may suffer in ODM. The crux of the issue here is that *only one arm can actually provide any new information* (i.e. the *request* arm). So the role of "exploration" here is very different: Bandit strategies would "explore" different arms pointlessly with no learning occurring most of the time. It is easy to see that this applies to all manner of algorithms such as $\epsilon$-greedy policies, posterior sampling, and strategies that rely on optimism in the face of uncertainty (the latter actually leading to strictly *fewer* request arms being pulled!). In Appendix C, we formally show that ODM is actually a distinct and concretely "harder" problem than contextual bandits (Definition 3)—precisely due to the nature of abstentive feedback.

## 3.2 UMPIRE Mediator Policy

Given the previous discussion, a simple "$\epsilon$-request" mediator policy may seem promising: It executes the greedy policy $\phi$, but with probability $\epsilon$ opts to request from the expert. Learning thus occurs more frequently, and there are no exploratory actions that yield no learning. But an important problem is that *not all exploratory actions are equally useful*: The value of any requested information surely depends on the context $x_t$; by randomizing "indiscriminately", an $\epsilon$-request strategy does not account for this.

We propose a more principled basis for interpolating between exploration and exploitation that we term *uncertainty-modulated policy for intervention and requisition* ("UMPIRE"). The main idea is that we might be willing to pay more to request an oracle action now, if it means we can more confidently rely on model predictions in the future (i.e. by accepting or intervening autonomously). So our motivation is to explicitly trade off (immediate) *system risk* with expected improvements in the (future) *model risk*.

To do so, we need a method that allows us to estimate the latter. Operating in the probabilistic setting, let $W$ denote the parameter variable, taking on values in $w \in \mathcal{W}$, such that we can write $\hat{\pi}_w(Y|X) := p(Y|X, w)$, and can also speak of the marginal $p(Y|D, X) = \mathbb{E}_{W \sim p(\cdot|D)} p(Y|X, W)$. Now, denote the *expected model risk* with $\bar{\mathcal{R}}(D) := \mathbb{E}_{W \sim p(\cdot|D)} \mathcal{R}(\hat{\pi}_W)$; note that this is itself a random variable due to its dependence on $D$. Consider the $t$-th round of play: If the mediator chooses $z \in \{0, 1\}$, then $d_t = d_{t-1}$ so we have $\bar{\mathcal{R}}(d_t) = \bar{\mathcal{R}}(d_{t-1})$. But in deciding whether to choose $z = 2$, we wish to capture a measure of how much this risk might possibly improve—that is, if we were to reveal (and learn from) the ground-truth label $Y_t$. So we are interested in how far $\bar{\mathcal{R}}(D_t)$ can end up relative to $\bar{\mathcal{R}}(d_{t-1})$, where $Y_t \sim p(\cdot|d_{t-1}, x_t)$. The following result gives such an upper bound on expected improvement:

**Theorem 1 (Expected Improvement)** Let $\mathcal{R}$ be bounded as $[-b, b]$—for instance, by centering $\ell_{01}$. Let $\mathbb{I}[W; Y_t|d_{t-1}, x_t]$ denote the mutual information between $W$ and $Y_t$ conditioned on $d_{t-1}$ and $x_t$, and let $W_0$ denote the principal branch of the product logarithm function. Then (proof in Appendix C):

$$\bar{\mathcal{R}}(d_{t-1}) - \mathbb{E}_{Y_t \sim p(\cdot|d_{t-1}, x_t)}[\bar{\mathcal{R}}(D_t)|d_{t-1}, x_t, Z_t = 2] \leq 2b(e^{W_0\left(\frac{1}{e}(\mathbb{I}[W;Y_t|d_{t-1},x_t]-1)\right)+1} - 1) \quad (6)$$

This motivates a straightforward technique: Define $g : v \mapsto g(v) = 2b(e^{W_0(\frac{1}{e}(v-1))+1} - 1)$, and let $\kappa$ denote some tradeoff coefficient. Then we can designate $\bar{k}_{\text{req}} := (1 - \kappa g(\mathbb{I}[W; Y_t|d_{t-1}, x_t]))k_{\text{req}}$, and simply use $\bar{k}_{\text{req}}$ in place of $k_{\text{req}}$ wherever it appears—but keeping the greedy mediator otherwise intact. UMPIRE thus has one hyperparameter $\kappa$; in our experiments we simply set its value as the normalizing constant $\kappa_0 := (2b(e^{W_0((\log m - 1)/e)+1} - 1))^{-1}$, which has the effect of keeping all costs non-negative.

**Interpretation** Since $g$ is monotonically increasing, Theorem 1 is naturally interpreted as translating an information-theoretic criterion (i.e. the mutual information) into a decision-theoretic criterion (i.e. the expected improvement in posterior risk)—which is what we require. In particular, the argument to $g$ expands as $\mathbb{I}[W; Y_t|d_{t-1}, x_t] = \mathbb{H}[W|d_{t-1}] - \mathbb{E}_{Y_t \sim p(\cdot|d_{t-1}, x_t)} \mathbb{H}[W|d_{t-1}, x_t, Y_t]$, which has the interpretation of how much the (epistemic) *uncertainty* in the model policy is expected to decrease if $Y_t \sim p(\cdot|d_{t-1}, x_t)$ is revealed; this view is reminiscent of entropy-based approaches to active learning [59, 60]. Observe that when deployed, $G_t := g(\mathbb{I}[W; Y_t|D_{t-1}, X_t])$ is large in the beginning, so $\bar{k}_{\text{req}}$ is small and UMPIRE behaves like standard incremental learning. In the limit of a perfect model, $G_t$ goes to zero, so $\bar{k}_{\text{req}} = k_{\text{req}}$ and UMPIRE behaves the same as the (optimal) greedy mediator $(\pi_*, \phi_*)$.

## 3.3 Practical Implementation

Some practical remarks deserve mention. First, UMPIRE is compatible with any choice of probabilistic modeling technique, such as Gaussian processes, Bayesian neural networks, and dropout-based approximations. Second, since integration over parameter posteriors is generally intractable, we use standard Monte-Carlo sampling to compute expectations: Let $s$ denote the number of samples taken from the posterior, and let $\{w_{i,t}\}_{i=1}^s$ indicate the set of samples drawn from $p(W|d_t)$. In computing the value of $g_t$, observe that the inner expression $\mathbb{E}_{Y_t \sim p(\cdot|d_{t-1}, x_t)} \mathbb{H}[W|d_{t-1}, x_t, Y_t]$ requires retraining the model policy on every possible value of $Y_t$. Instead, we can rely on the symmetry of mutual information and expand $\mathbb{I}[Y_t; W|d_{t-1}, x_t] = \mathbb{H}[Y_t|d_{t-1}, x_t] - \mathbb{E}_{W \sim p(\cdot|d_{t-1})} \mathbb{H}[Y_t|x_t, W]$, so we can write:

$$\hat{\mathbb{I}}[W; Y_t|d_{t-1}, x_t] := H[\tfrac{1}{s} \textstyle\sum_{i=1}^s p(Y_t|x_t, w_{i,t-1})] - \tfrac{1}{s} \textstyle\sum_{i=1}^s H[p(Y_t|x_t, w_{i,t-1})] \quad (7)$$

where we define $H[p(Y)] := -\sum_{y \in \mathcal{Y}} p(y) \log p(y)$, giving us a more efficient way to compute the expression without such retraining (see Appendix C for detail). Lastly, to guarantee consistency we can easily still request from the expert with some small probability $\epsilon$ (i.e. in the same way the "$\epsilon$-request" policy above does so over the greedy policy). Algorithm 1 summarizes UMPIRE as applied to ODM.

**Algorithm 1** UMPIRE Mediator  ▷ for Online Decision Mediation
___
1: **Hyperparameters**: tradeoff coefficient $\kappa$, Monte-Carlo samples $s$
2: **Input**: initial dataset $d_0$, cost of intervention $k_{\text{int}}$, cost of requisition $k_{\text{req}}$
3: **for** each round $t = 1, ...$ **do**
4:    $x_t \leftarrow X_t \sim \rho$
5:    $\tilde{y}_t \leftarrow \tilde{Y}_t \sim \tilde{\pi}(\cdot|x_t)$  ▷ human action
6:    $\hat{\pi}(Y|x_t) := \frac{1}{s}\sum_{i=1}^{s} p(Y|x_t, w_{i,t-1})$
7:    $\hat{y}_t \leftarrow \arg\max_y \hat{\pi}(y|x_t)$  ▷ model action
8:    $\hat{\mathbb{I}}[W; Y_t|d_{t-1}, x_t] \leftarrow H[\frac{1}{s}\sum_{i=1}^{s} p(Y_t|x_t, w_{i,t-1})] - \frac{1}{s}\sum_{i=1}^{s} H[p(Y_t|x_t, w_{i,t-1})]$
9:    $\phi(Z|x_t, \tilde{y}_t) := \delta(Z - \arg\min_z (\mathbb{1}_{[z=0]}(1 - \hat{\pi}(\tilde{y}_t|x_t)) + \mathbb{1}_{[z=1]}(1 - \hat{\pi}(\hat{y}_t|x_t) + k_{\text{int}})$
10:   $z_t \leftarrow Z_t \sim \phi(\cdot|x_t, \tilde{y}_t)$ $\qquad\qquad + \mathbb{1}_{[z=2]}(1 - \kappa g(\hat{\mathbb{I}}[W; Y_t|d_{t-1}, x_t]))k_{\text{req}}))$
11:   **if** $z_t = 2$ **then**
12:       $y_t \leftarrow Y_t \sim \pi_*(\cdot|x_t)$  ▷ expert action
13:       $d_t \leftarrow d_{t-1} \cup \{(x_t, y_t)\}$
14:   **else** $d_t \leftarrow d_{t-1}$
15:   **Output**: $\bar{y}_t \leftarrow \mathbb{1}_{[z_t=0]}\tilde{y}_t + \mathbb{1}_{[z_t=1]}\hat{y}_t + \mathbb{1}_{[z_t=2]}y_t$  ▷ (final) system action

# 4 Empirical Results

Three aspects of UMPIRE deserve investigation: **(a) Performance**: *Does it work?* Section 4.1 compares it to existing methods, validating its role in decision support by most consistently improving decisions. **(b) Source of Gain**: *Why does it work?* Section 4.2 deconstructs the key characteristics of UMPIRE, verifying the importance of each. **(c) Sensitivity Analysis**: Finally, Section 4.3 assesses the sensitivity of UMPIRE and benchmarks to the expert's stochasticity, costs of request, and number of samples used.

**Datasets** We experiment with six environments. In `GaussSine`, synthetic points are generated in three categories by rounding a sinusoidal latent function on 2D Gaussian input [61]. In `HighEnergy`, the task is to identify signals in high energy particles registered in a Cherenkov gamma telescope [62]. In `MotionCapture`, the task is to recognize hand postures from data recorded by glove markers on users [63]. In `LunarLander`, the task is to perform actions in the OpenAI gym [64] Atari environment, with the expert defined as a PPO2 agent [65,66] trained on the true reward. In `Alzheimers`, the task is to perform early diagnosis of patients in the Alzheimer's Disease Neuroimaging Initiative study [67] as cognitively normal, mildly impaired, or at risk of dementia [19,20]. Lastly, in `CysticFibrosis`, the task is to perform diagnosis of patients enrolled in the UK Cystic Fibrosis registry [68] as to their GOLD grading in chronic obstructive pulmonary disease [69]. See Appendix B for additional detail.

**Benchmarks** We consider adaptations of algorithms from related work. First as our minimal baseline, `Human` always accepts $z = 0$, thus constituting the starting point for performance comparison. `Random` draws $z$ at random. `Supervised` picks $z \in \{0, 1\}$ based solely on $\hat{\pi}$'s output, and $z = 2$ w.p. $\epsilon$, and thus resembles supervised learning. `Cost-Sensitive` is the greedy $(\hat{\pi}, \phi_*)$ from Section 3, which additionally accounts for costs $k_{\text{int}}, k_{\text{req}}$. `Thompson Sampling` [46] draws from the posterior $p(W|d)$ and selects $z$ greedily using $(\hat{\pi}_W, \phi_*)$; the `Full` version also uses that sampled model when predicting. `Epsilon-Greedy` [47] is greedy but draws $z$ at random w.p. $\epsilon$; the `Request` version is the smarter "$\epsilon$-request" from Section 3.2. `Pessimistic Bayesian Sampling` adapts OBS [44] to ODM by reversing the direction of optimism such that the tendency to request actually *increases*. `Bayesian Active Request` adapts Bayesian active learning [60] to ODM by requesting w.p. $\sim$ expected reduction in entropy. To further highlight the advantage of our proposed criterion, `Matched Decaying Request` is an artificially boosted benchmark that is similar to $\epsilon$-request—but where $\epsilon$ is a decay function that has the benefit of matching the effective request rate of UMPIRE: This is done *post-hoc* by searching for a polynomial function that best models UMPIRE's request pattern. See Appendix B for additional detail.

**Experiment Setup** Each experiment run consists of $n = 2000$ rounds of interactions (except for the synthetic GaussSine, for which $n = 500$), and this is repeated for a total of 10 runs with random seeds. For all algorithms, the underlying model policy is implemented identically using Dirichlet-based Gaussian process classifiers [61,70–72]. We simulate fallible human decisions as random perturbations of the ground truth with some probability $\alpha$. As mentioned in Section 2.1, we let $k_{\text{req}} = \frac{m}{m-1} - \gamma$ for some small $\gamma > 0$: To do so, we simply set $k_{\text{req}}$ to $\frac{m}{m-1}$ rounded down to the nearest decimal point. However, we shall perform additional sensitivities on this below. In all experiments, we set $k_{\text{int}} = 0.1$, $\alpha = \frac{1}{2}$, $\epsilon = 10\%$ where applicable, and $\kappa = \kappa_0$ as noted in Section 3.2. Regret is defined with respect to the oracle mediator $(\pi_*, \phi_*)$, for which $\pi_*$ is approximated by training on the full dataset in advance. Performance metrics for each benchmark are reported as means and standard deviations across all runs.

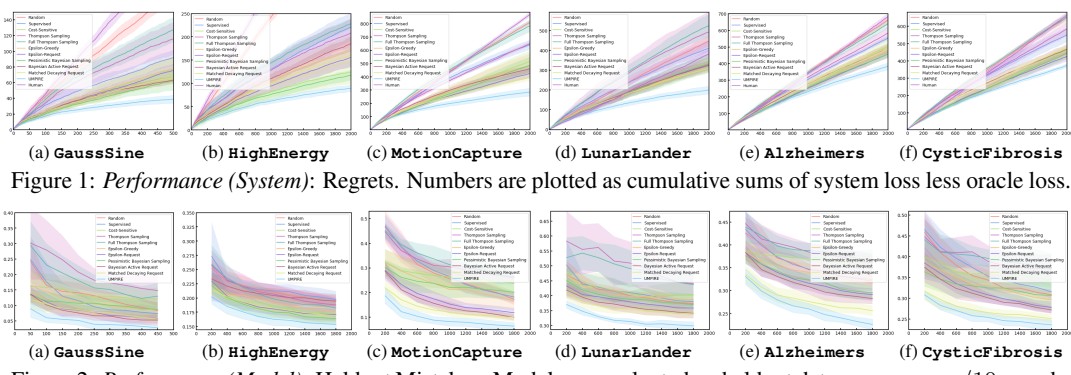

(a) `GaussSine`    (b) `HighEnergy`    (c) `MotionCapture`    (d) `LunarLander`    (e) `Alzheimers`    (f) `CysticFibrosis`

Figure 1: *Performance (System)*: Regrets. Numbers are plotted as cumulative sums of system loss less oracle loss.

(a) `GaussSine`    (b) `HighEnergy`    (c) `MotionCapture`    (d) `LunarLander`    (e) `Alzheimers`    (f) `CysticFibrosis`

Figure 2: *Performance (Model)*: Heldout Mistakes. Models are evaluated on heldout data once every $n/10$ rounds.

| Benchmark | GaussSine Err. Acc. | Exc. Int. | Abs. Shf. | HighEnergy Err. Acc. | Exc. Int. | Abs. Shf. | MotionCapture Err. Acc. | Exc. Int. | Abs. Shf. | LunarLander Err. Acc. | Exc. Int. | Abs. Shf. | Alzheimers Err. Acc. | Exc. Int. | Abs. Shf. | CysticFibrosis Err. Acc. | Exc. Int. | Abs. Shf. |
|---|---|---|---|---|---|---|---|---|---|---|---|---|---|---|---|---|---|---|
| Random | 116±8 | 131±7 | 37±8 | 452±17 | 545±19 | 199±11 | 454±18 | 572±24 | 260±21 | 449±18 | 634±21 | 386±26 | 448±22 | 615±30 | 326±22 | 448±22 | 616±23 | 325±20 |
| Supervised | 19±8 | 47±7 | 34±7 | 204±17 | 201±20 | 203±18 | 39±8 | 485±34 | 256±14 | 114±18 | 661±39 | 388±27 | 201±29 | 466±22 | 330±14 | 155±23 | 536±46 | 344±22 |
| Cost-Sensitive | 33±14 | 36±11 | 37±11 | 176±15 | 139±16 | 176±15 | 101±36 | 352±48 | 232±41 | 166±20 | 607±36 | 403±18 | 305±29 | 333±21 | 333±21 | 265±16 | 299±19 | 296±11 |
| Thompson Sampling | 44±10 | 90±18 | 50±10 | 229±24 | 226±17 | 194±20 | 118±14 | 622±18 | 303±10 | 202±39 | 882±79 | 502±63 | 275±20 | 603±19 | 379±24 | 284±13 | 628±32 | 374±21 |
| Full Thompson Sampling | 41±7 | 105±14 | 64±7 | 231±10 | 252±14 | 219±9 | 119±18 | 728±32 | 415±28 | 205±28 | 907±43 | 552±31 | 277±24 | 665±35 | 430±17 | 273±24 | 714±33 | 450±13 |
| Epsilon-Greedy | 29±9 | 42±11 | 28±11 | 196±20 | 177±28 | 174±20 | 95±28 | 319±66 | 194±47 | 172±19 | 587±24 | 385±21 | 260±31 | 337±21 | 290±20 | 226±24 | 315±23 | 267±19 |
| Epsilon-Request | 16±7 | 25±7 | 22±4 | 162±13 | 124±14 | 162±13 | 32±6 | 220±31 | 132±15 | 129±19 | 523±42 | 342±26 | 220±21 | 284±13 | 265±15 | 167±22 | 275±27 | 234±12 |
| Pessimistic Bayesian Sampling | 16±7 | 45±9 | 26±7 | 139±14 | 147±10 | 143±14 | 41±19 | 441±39 | 194±31 | 111±18 | 704±47 | 364±30 | 177±22 | 449±25 | 275±17 | 136±19 | 429±22 | 248±21 |
| Bayesian Active Request | 12±6 | 25±7 | 19±4 | 191±14 | 192±18 | 192±13 | 19±6 | 232±26 | 125±14 | 87±11 | 532±26 | 304±19 | 143±16 | 351±18 | 238±16 | 114±26 | 371±24 | 232±20 |
| Matched Decaying Request | 15±6 | 23±8 | 20±4 | 154±11 | 122±12 | 154±11 | 21±4 | 158±23 | 93±13 | 102±12 | 449±23 | 291±16 | 128±12 | 211±17 | 175±14 | 93±11 | 198±21 | 155±12 |
| **UMPIRE** | 4±3 | 6±3 | 6±4 | 112±15 | 86±9 | 112±15 | 13±5 | 67±28 | 42±17 | 64±12 | 343±16 | 212±15 | 90±12 | 130±18 | 125±11 | 80±13 | 132±12 | 119±15 |

Table 2: *Performance (Mediator)*: Erroneous acceptance, excessive intervention, and abstention shortfall at $t = n$.

## 4.1 Performance

Recall from Section 1 we motivated ODM from the view of *decision support*: To best improve decision-making by the (human-expert-mediator) system as a whole. Primarily, then, we evaluate the performance of the *decision system*—that is, in terms of system regret (Equation 4). Figure 1 shows results for all algorithms, with `Human` (i.e. without support) also shown for comparison: UMPIRE consistently accumulates lower regret. This is our objective function, and this is the main takeaway. Secondarily, we can also assess the (heldout) performance of the *model policy*—that is, if it were tasked with acting autonomously at any point. Figure 2 shows the rate of heldout mistakes: UMPIRE appears to induce more efficient learning. As another auxiliary, we can also consider how the *mediator policy* behaves—that is, if it accepts erroneously ($z = 0$ but $\tilde{y} \neq y$), intervenes excessively ($z = 1$ but oracle $z_* \in \{0, 2\}$), or abstains not conservatively enough ($z \in \{0, 1\}$ while $\tilde{y} \neq y$ and $\hat{y} \neq y$). Table 2 shows the results, likewise with UMPIRE as best. (Note that this is not simply due to more frequent requests, since `Matched Decaying Request` selects $z = 2$ just as often as UMPIRE using a dynamic $\epsilon_t$ tuned post-hoc). For more comprehensive measures of the system (loss, regret, mistakes) and model (heldout cross entropy, mistakes, AUROC, and AUPRC), see Figures 8, 9, 10, 11, 12, 13, and 14 in Appendix A.

## 4.2 Source of Gain

Recall from Section 2 that ODM combines the challenges from all three related settings, and from Section 3 that UMPIRE is designed with three corresponding desiderata in mind. We now examine each aspect's contribution to final performance. Table 3 enumerates some settings to isolate the following: (i) Does it act exploit $\hat{\pi}$ in a cost-sensitive manner ("CS"), like in learning with rejection? (ii) Does it attempt to explore deliberately in addition ("DE"), like in bandit

| Setting | CS | DE | UA |
|---|---|---|---|
| No Request | ✗ | ✗ | ✗ |
| Passive Request | ✓ | ✗ | ✗ |
| Epsilon-Request w/o CS | ✗ | ✓ | ✗ |
| Epsilon-Request with CS | ✓ | ✓ | ✗ |
| Dynamic-Request w.p. KG | ✗ | ✓ | ✓ |
| Dynamic-Request with ME | ✓ | ✓ | ✗ |
| **UMPIRE** | ✓ | ✓ | ✓ |

Table 3: *Source-of-Gain Legend.*

algorithms? (iii) Does it leverage uncertainty awareness in decision-making ("UA"), similar to active learning? (Abbreviations: `Dynamic-Request w.p. KG` requests w.p. $\kappa G_t$, and `Dynamic-Request with ME` requests with matched $\epsilon_t$, i.e. same as `Matched Decaying Request` from above). Figure 3 shows results for the decision system in terms of system regret, and secondarily Figure 4 shows results for the learned model in terms of heldout mistakes: It is apparent that all three aspects are crucial for performance, and cannot be neglected. For more comprehensive source-of-gain evaluation metrics for the system, model, and mediator, see Figures 15, 16, 17, 18, 19, 20, and 21, and Table 5 in Appendix A.

## 4.3 Sensitivity Analysis

Lastly, we examine UMPIRE's sensitivity to several factors. First, the expert might be more or less stochastic depending on the environment. We simulate this by progressively injecting additional noise to the latent function in `GaussSine`. Figure 5 shows the results: The advantage of UMPIRE is most

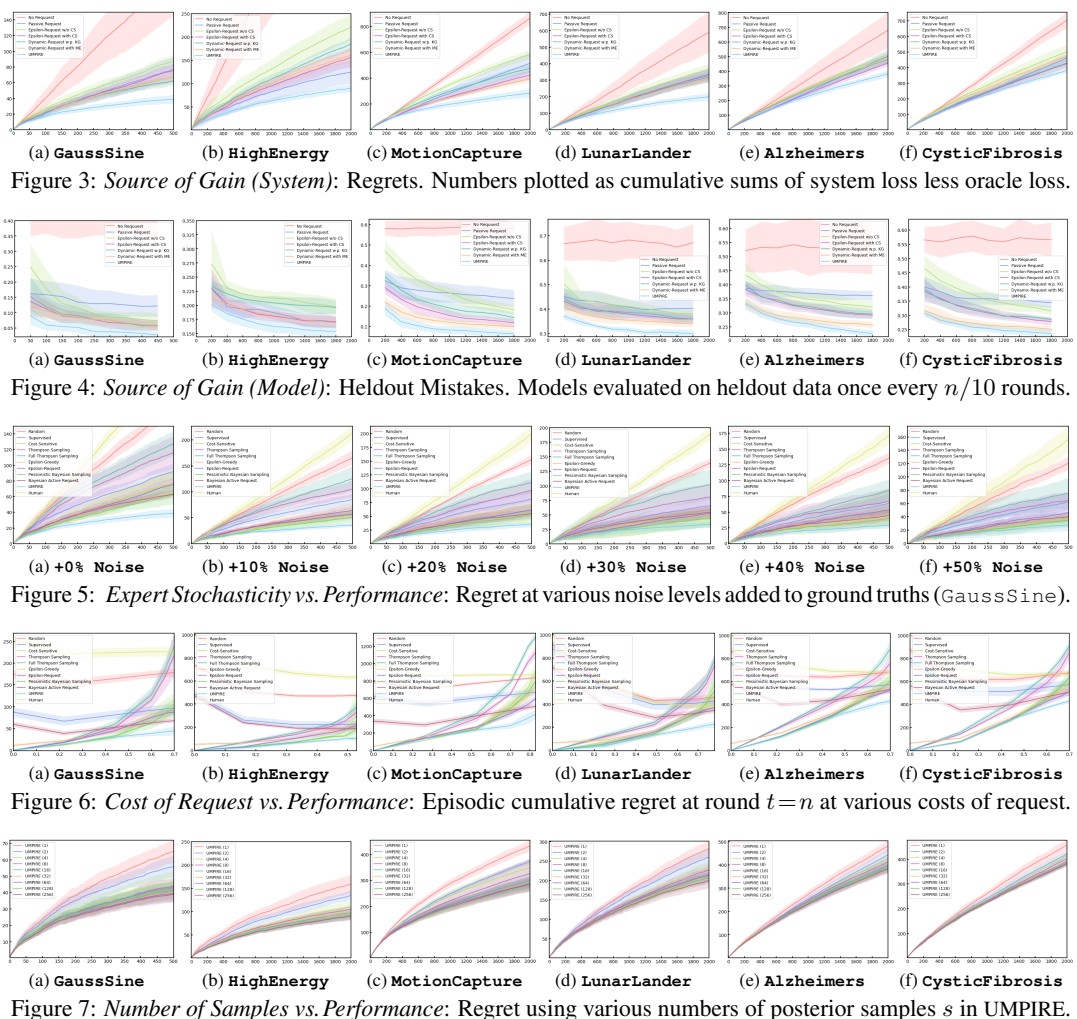

Figure 3: *Source of Gain (System)*: Regrets. Numbers plotted as cumulative sums of system loss less oracle loss.

| (a) GaussSine | (b) HighEnergy | (c) MotionCapture | (d) LunarLander | (e) Alzheimers | (f) CysticFibrosis |

Figure 4: *Source of Gain (Model)*: Heldout Mistakes. Models evaluated on heldout data once every $n/10$ rounds.

| (a) GaussSine | (b) HighEnergy | (c) MotionCapture | (d) LunarLander | (e) Alzheimers | (f) CysticFibrosis |

Figure 5: *Expert Stochasticity vs. Performance*: Regret at various noise levels added to ground truths (GaussSine).

| (a) +0% Noise | (b) +10% Noise | (c) +20% Noise | (d) +30% Noise | (e) +40% Noise | (f) +50% Noise |

Figure 6: *Cost of Request vs. Performance*: Episodic cumulative regret at round $t = n$ at various costs of request.

| (a) GaussSine | (b) HighEnergy | (c) MotionCapture | (d) LunarLander | (e) Alzheimers | (f) CysticFibrosis |

Figure 7: *Number of Samples vs. Performance*: Regret using various numbers of posterior samples $s$ in UMPIRE.

pronounced at lower noise levels, and decreases as noise levels rise; this makes sense, as our estimates of uncertainty become more entangled with noise. For more comprehensive results, see Figures 22, 23, 24, and 25 in Appendix A. Second, we examine the consequence of different costs of request. Recall that we have been operating in the regime where $k_{\text{req}} = \frac{m}{m-1} - \gamma$ for some small $\gamma > 0$ [22,25,37,40]. We now allow the cost of request to vary from $k_{\text{req}} = 0$ to $\frac{m}{m-1}$ as sensitivities. (For completeness, we actually go slightly beyond the standard threshold and go as high as $\gamma = -0.05 < 0$.) Figure 6 shows the results: The advantage of UMPIRE appears most pronounced at higher costs, and decreases in the opposite direction; this makes sense, as cheaper costs mean abstention becomes a more trivial decision. For more comprehensive results, see Figures 26, 27, 28, and 29 in Appendix A. Finally, recall that Equation 7 requires estimating the mutual information by Monte-Carlo samples. Figure 7 shows the sensitivity of performance to the practical choice of sample size $s$: Observe that performance appears to stabilize with reasonable choices of $s \approx 64$ and above. See Appendix A for additional detail.

**Additional Sensitivities** In the above experiments, the imperfect human decisions are simulated with $\alpha = \frac{1}{2}$. However, it should be clear that the specific pattern or frequency of mistakes does not affect the basic structure of the problem, as long as the imperfect decisions stochastically deviate from the expert's with some probability between zero and one. Appendices E.3–E.4 perform a complete re-run of all experiments under the same conditions as before—but now setting $\alpha = \frac{9}{10}$. Similarly, we can verify that our intuitions still hold when the cost of intervention $k_{\text{int}} = 0.0$, which corresponds to the special case where the machine behaves "autonomously" for all intents and purposes, except where it requests input from the expert. Appendices E.5–E.6 perform experiments for the cartesian product of settings $\alpha \in \{0.5, 0.7, 0.9, 1.0\}$ and $k_{\text{int}} \in \{0.0, 0.1\}$, using the GaussSine environment. Across all sensitivities, it is easy to see that UMPIRE still consistently accumulates lower regret (our primary metric of interest), as well as outperforming comparators with respect to to the rest of the performance measures.

# 5 Discussion

Research in machine learning for decision-making is proliferating, such as in data-driven clinical decision support—but much focus is exclusively placed on comparing computers *versus* clinicians [73]: Less explored is how machines can serve as adjuncts to make decision systems more efficient *as a unit*. In this work, we take a first step by formalizing ODM as a sequential problem, proposing UMPIRE as a potential solution, and demonstrating the importance of considering all aspects of this unique setting. While this perspective is general, the ethical responsibility for a decision—e.g. signing off on a diagnosis—often must ultimately fall on humans: To remain vigilant of potential bias in societal impact, it is thus crucial to examine the complementary problem of "closing the loop" by considering how humans themselves may in turn interpret feedback to modify their own behavior [74]. Moreover, future work may explore generalizing ODM and its solutions—such as to settings with differential costs of mistake or class imbalances, or to consider aspects of interpretability in model policies for human feedback.

**Applications** In addition to clinical decision support, the ODM problem setting is applicable to any scenario where "imperfect" decision-makers are the front-line decision-makers, and "oracle" decision-makers are available as expert supervision—albeit with limited availability, and where learning feedback is abstentive. This situation arises in many settings where the responsibility for a decision must ultimately fall on a *person* (i.e. the imperfect human or the expert), but a *machine* is available for learning and issuing recommendations. The following are some potential examples of such:

- *Product Inspection*: Suppose a junior employee signs off on the quality of a product batch. The mediator can decide to (1) accept the sign-off *as is*, or (2) recommend a re-inspection for the batch, due to a disagreeing autonomous prediction as to the product quality, or (3) recommend that a more senior employee take over and issue their more qualified assessment.

- *Content Moderation*: Suppose users in a social network can report suspected content violations in real time. The mediator can decide to (1) accept and act on a user's report *as is*, or (2) recommend that the user re-classify the content due to a disagreeing assessment as to its appropriateness, or (3) recommend that an internal moderator take over and issue their judgement.

- *Spoken-Dialog System*: Suppose a customer selects a possibly-nonsensical option in a spoken-dialog system. The mediator can decide to (1) accept and act on the user's option *as is*, or (2) recommend that the user re-select an option from the same menu, or (3) re-route the customer to a phone conversation with an actual (human) customer representative to continue the work.

Finally, note that ODM is also applicable in settings where there is no imperfect human involved, so the machine simply makes decisions autonomously and learns from selective expert feedback; this is simply the setting where $k_{\text{int}} = 0$. Importantly, however, this does not alter the basic structure of the ODM problem, whose hardness is distinguished by the fact that expert feedback is costly and abstentive.

**Limitations** There are two main limitations of our analysis: First, ODM is an *online learning* framework. In general, it is known that online learning may not perform well during early time steps when the learner's decisions are largely exploratory, especially if learning proceeds "from scratch"—which is the setting we operate in. In this sense, UMPIRE as a solution is also not immune to this challenge. Therefore, future work would benefit from examining the potential to *not* learn from scratch—e.g. to "warm-start" a learner using existing data, which can be done using a variety of methods from the extensive literature on imitation learning. Second, we must recognize that there are *two sides* to human-machine interactions: While ODM focuses on how machines should best propose recommendations to humans, there is also the complementary aspect of how/whether humans actually incorporate such recommendations into their behavior. Ignoring this second aspect may lead to models that are accurate but not necessarily best at proposing recommendations that are most likely to be complied with—which would severely undermine the practical utility of such a model. Therefore, future work would also benefit from *jointly* studying how humans and machines should behave in a "mutually-aware" fashion.

## Acknowledgments

We would like to thank the reviewers for their comments, suggestions, and generous feedback. This work was supported by Alzheimer's Research UK, The Alan Turing Institute under EPSRC grant number EP/N510129/1, the United States Office of Naval Research (ONR), as well as the National Science Foundation (NSF) under grant numbers 1407712, 1462245, 1524417, 1533983, and 1722516.

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
