# A Additional Results

Sections A.1–A.3 present more comprehensive empirical results organized to correspond with Sections 4.1–4.3. The objective of ODM is to minimize the *system regret*, so that should be the primary measure of performance; however, we also provide a variety of other metrics to give a broader picture of how UMPIRE and benchmarks behave. In order of appearance in the sequel: The **system loss** is defined by Equation 3: $\mathcal{R}_t(\hat{\pi}, \phi) := \mathbb{E}_{X_t \sim \rho} \mathbb{E}_{\tilde{Y}_t \sim \tilde{\pi}(\cdot|X_t)} \mathbb{E}_{Y_t \sim \pi_*(\cdot|X_t)} [\phi(Z_t = 0|X_t, \tilde{Y}_t) \ell(Y_t, \delta(Y - \tilde{Y}_t)) + \phi(Z_t = 1|X_t, \tilde{Y}_t)(\ell(Y_t, \hat{\pi}(\cdot|X_t)) + k_{\text{int}}) + \phi(Z_t = 2|X_t, \tilde{Y}_t) k_{\text{req}}]$, where we approximate the expectations over multiple samples by averaging over the ten runs of each experiment; loss values are taken as the moving average of a rolling window of width $n/5$. The **system regret** is defined by Equation 4: $\text{Regret}[t] := \sum_{\tau=0}^{t} (\mathcal{R}_\tau(\hat{\pi}, \phi) - \mathcal{R}_\tau(\pi_*, \phi_*))$; that is, regret values are the cumulative sums of system loss less oracle loss (with the oracle mediator policy being the greedy policy $\phi_*$ and the oracle model policy $\pi_*$ approximated by training on the full dataset in advance), where we similarly approximate the expectations over multiple trajectories by averaging over the ten runs of each experiment. The **system mistake** at each time $t$ is defined as whether or not the decision system $\mathcal{S}$ as a whole outputs a final decision $\bar{Y}_t \sim \pi_{\mathcal{S}}(\cdot|X, \tilde{Y})$ that is a mistake; that is, if $\bar{Y}_t$ is different from the (observed or unobserved) decision of the expert $Y_t$: $\mathbb{E}_{X_t \sim \rho} \mathbb{E}_{\tilde{Y}_t \sim \tilde{\pi}(\cdot|X_t)} \mathbb{E}_{Y_t \sim \pi_*(\cdot|X_t)} \ell(Y_t, \bar{\pi}_{\mathcal{S}}(\cdot|X_t, \tilde{Y}_t))$, where we again likewise approximate the expectations over multiple samples by averaging over the ten runs of each experiment; as with system loss, numbers are computed as the moving average of a rolling window of width $n/5$. Across all evaluation measures, UMPIRE consistently performs better than applying existing methods.

Secondarily, heldout metrics measure how well the learned model policy $\hat{\pi}$ would do if asked to make decisions $\hat{Y} \sim \hat{\pi}(\cdot|X)$ autonomously. These compute the *model risk* as in Equation 2: $\text{Model-Risk}[t] := \mathbb{E}_{X \sim \rho} \mathbb{E}_{Y \sim \pi_*(\cdot|X)} \ell(Y, \hat{\pi}_t(\cdot|X))$ where $\hat{\pi}_t$ is the model policy at the end of round $t$ (i.e. learned on the dataset $D_t := \{(X_\tau, Y_\tau) : Z_\tau = 2\}_{\tau=1}^{t}$), where we approximate the expectations by averaging over heldout datasets of size $n = 2000$. This is computed once every $n/10$ rounds to show its evolution across $t$, and a different heldout dataset is randomly drawn for every run. The **heldout mistakes** metric uses the zero-one loss; the **heldout cross entropy** uses the cross entropy between expert and model policies; the **heldout AUROC** refers to the area under the receiver operating characteristic curve; and the **heldout AUPRC** refers to the area under the precision-recall curve. Lastly, measures of the mediator policy include the cumulative numbers of erroneous acceptances, excessive interventions, and abstention shortfalls throughout each trajectory, as defined as in Section 4.1, again averaged over ten runs. As before, across all measures, UMPIRE consistently performs better than applying existing methods.

Section A.2 presents the **source-of-gain** analysis, which follows a similar format to Section A.1, but now comparing UMPIRE against different versions of itself—with various (combinations of) characteristics removed (viz. Table 3), incl. cost-sensitivity, deliberate exploration, and uncertainty awareness. As noted in the manuscript, it is apparent that all three aspects of UMPIRE are crucial for performance.

Section A.3 presents various sensitivity analyses, likewise following a similar format to Section 4.3 but including additional results. First, we look at the effect of **expert stochasticity** on the performance of UMPIRE and benchmarks: This is accomplished by injecting additional noise to the latent function in GaussSine: Since ground-truth labels are generated as $y = \text{round}(f(x_1, x_2))$ for some latent function $f$ (see Section B), we add additional noise by setting $y = \text{round}(f(x_1, x_2) + \text{uniform}(-\frac{q}{2}, \frac{q}{2}))$ for $q \in \{0.0, 0.1, 0.2, 0.3, 0.4, 0.5\}$. Results for sensitivities are computed and presented in relation with benchmark comparators, as well as with source-of-gain comparators. In both cases, the advantage of UMPIRE appears the most pronounced at lower noise levels, and decreases with higher noise levels.

Second, we look at the effect of **cost of request** on performance: This is achieved by executing the entire experiment procedure at various values of $k_{\text{req}}$. As above, results are presented in relation to benchmark comparators and source-of-gain comparators. As noted in the manuscript, in this work we focus on the most common regime where $k_{\text{req}} = \frac{m}{m-1} - \gamma$ for some $\gamma > 0$ [22,23,25,37,40]; as in [25], we refer to [22] for a discussion on how the case $\gamma \leq 0$ yields a fundamentally different class of problems. (However, for completeness we do go beyond the cutoff and experiment with up to $\gamma = -0.05 < 0$.) To show how results change across the range of values for $k_{\text{req}}$, we report the *episodic average loss* (across all $n$ rounds) versus the cost of request, as well as the (cumulative) *system regret* (at round $t = n$) versus the cost of request. Both metrics are reported for both benchmark comparators and source-of-gain comparators. The advantage of UMPIRE is the most pronounced at higher costs (where the passive exploration induced by the greedy policy plays a smaller role), and decreases in the opposite direction.

Finally, in all experiments so far we used $s = 256$ **Monte-Carlo samples** in our UMPIRE implementation. We can look at the effect of $s$: Performance appears to be reasonably good at $s \approx 64$ and above.

## A.1 Performance

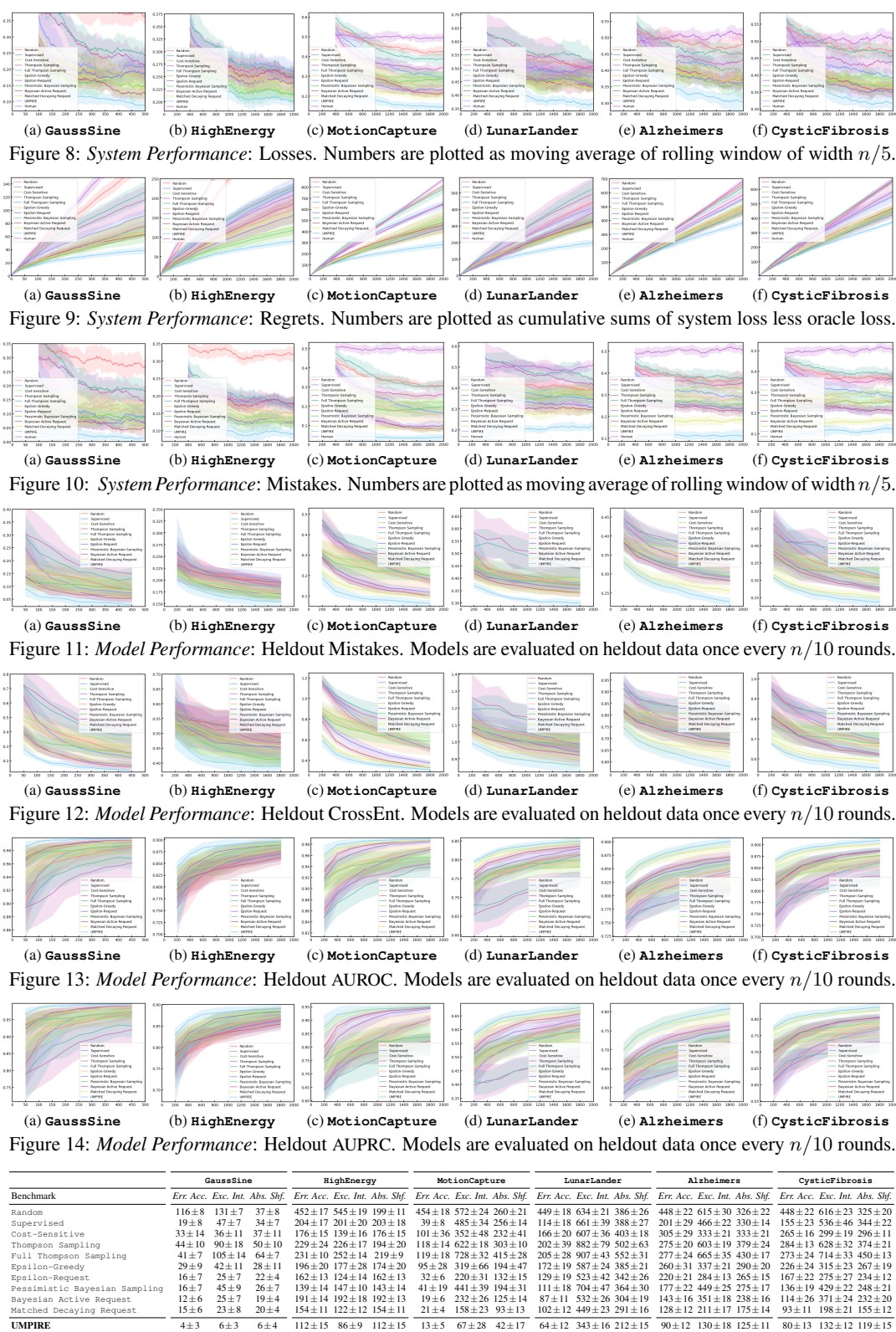

Figure 8: *System Performance*: Losses. Numbers are plotted as moving average of rolling window of width $n/5$.

(a) GaussSine (b) HighEnergy (c) MotionCapture (d) LunarLander (e) Alzheimers (f) CysticFibrosis

Figure 9: *System Performance*: Regrets. Numbers are plotted as cumulative sums of system loss less oracle loss.

(a) GaussSine (b) HighEnergy (c) MotionCapture (d) LunarLander (e) Alzheimers (f) CysticFibrosis

Figure 10: *System Performance*: Mistakes. Numbers are plotted as moving average of rolling window of width $n/5$.

(a) GaussSine (b) HighEnergy (c) MotionCapture (d) LunarLander (e) Alzheimers (f) CysticFibrosis

Figure 11: *Model Performance*: Heldout Mistakes. Models are evaluated on heldout data once every $n/10$ rounds.

(a) GaussSine (b) HighEnergy (c) MotionCapture (d) LunarLander (e) Alzheimers (f) CysticFibrosis

Figure 12: *Model Performance*: Heldout CrossEnt. Models are evaluated on heldout data once every $n/10$ rounds.

(a) GaussSine (b) HighEnergy (c) MotionCapture (d) LunarLander (e) Alzheimers (f) CysticFibrosis

Figure 13: *Model Performance*: Heldout AUROC. Models are evaluated on heldout data once every $n/10$ rounds.

(a) GaussSine (b) HighEnergy (c) MotionCapture (d) LunarLander (e) Alzheimers (f) CysticFibrosis

Figure 14: *Model Performance*: Heldout AUPRC. Models are evaluated on heldout data once every $n/10$ rounds.

| Benchmark | GaussSine | | | HighEnergy | | | MotionCapture | | | LunarLander | | | Alzheimers | | | CysticFibrosis | | |
| --- | --- | --- | --- | --- | --- | --- | --- | --- | --- | --- | --- | --- | --- | --- | --- | --- | --- | --- |
| | Err. Acc. | Exc. Int. | Abs. Shf. | Err. Acc. | Exc. Int. | Abs. Shf. | Err. Acc. | Exc. Int. | Abs. Shf. | Err. Acc. | Exc. Int. | Abs. Shf. | Err. Acc. | Exc. Int. | Abs. Shf. | Err. Acc. | Exc. Int. | Abs. Shf. |
| Random | 116±8 | 131±7 | 37±8 | 452±17 | 545±19 | 199±11 | 454±18 | 572±24 | 260±21 | 449±18 | 634±21 | 386±26 | 448±22 | 615±30 | 326±22 | 448±22 | 616±23 | 325±20 |
| Supervised | 19±8 | 47±7 | 34±7 | 204±17 | 201±20 | 203±18 | 39±8 | 485±34 | 256±14 | 114±18 | 661±39 | 388±27 | 201±29 | 466±22 | 330±14 | 155±23 | 536±46 | 344±22 |
| Cost-Sensitive | 33±14 | 36±11 | 37±11 | 176±15 | 139±16 | 176±15 | 101±36 | 352±48 | 232±41 | 166±20 | 607±36 | 403±18 | 305±29 | 333±21 | 333±21 | 265±16 | 299±19 | 296±11 |
| Thompson Sampling | 44±10 | 90±18 | 50±10 | 229±24 | 226±17 | 194±20 | 118±14 | 622±18 | 303±10 | 202±39 | 882±79 | 502±63 | 275±20 | 603±19 | 379±24 | 284±13 | 628±32 | 374±21 |
| Full Thompson Sampling | 41±7 | 105±14 | 64±7 | 231±10 | 252±14 | 219±9 | 119±18 | 728±32 | 415±28 | 205±28 | 907±43 | 552±31 | 277±24 | 665±35 | 430±17 | 273±24 | 714±33 | 450±13 |
| Epsilon-Greedy | 29±9 | 42±11 | 28±11 | 196±20 | 177±28 | 174±20 | 95±28 | 319±66 | 194±47 | 172±19 | 587±24 | 385±21 | 260±31 | 337±21 | 290±20 | 226±24 | 315±23 | 267±19 |
| Epsilon-Request | 16±7 | 25±7 | 22±4 | 162±13 | 124±14 | 162±13 | 32±6 | 220±31 | 132±15 | 129±19 | 523±42 | 342±26 | 220±21 | 284±13 | 265±15 | 167±22 | 275±27 | 234±12 |
| Pessimistic Bayesian Sampling | 16±7 | 45±9 | 26±7 | 139±14 | 147±10 | 143±14 | 41±19 | 441±39 | 194±31 | 111±18 | 704±47 | 364±30 | 177±22 | 449±25 | 275±17 | 136±19 | 429±22 | 248±21 |
| Bayesian Active Request | 12±6 | 25±7 | 19±4 | 191±14 | 192±18 | 192±13 | 19±6 | 232±26 | 125±14 | 87±11 | 532±26 | 304±19 | 143±16 | 351±18 | 238±16 | 114±26 | 371±24 | 232±20 |
| Matched Decaying Request | 15±6 | 23±8 | 20±4 | 154±11 | 122±12 | 154±11 | 21±4 | 158±23 | 93±13 | 102±12 | 449±23 | 291±16 | 128±12 | 211±17 | 175±14 | 93±11 | 198±21 | 155±12 |
| **UMPIRE** | 4±3 | 6±3 | 6±4 | 112±15 | 86±9 | 112±15 | 13±5 | 67±28 | 42±17 | 64±12 | 343±16 | 212±15 | 90±12 | 130±18 | 125±11 | 80±13 | 132±12 | 119±15 |

Table 4: *Mediator Performance*: Erroneous acceptance, excessive intervention, and abstention shortfall at $t = n$.

## A.2 Source of Gain

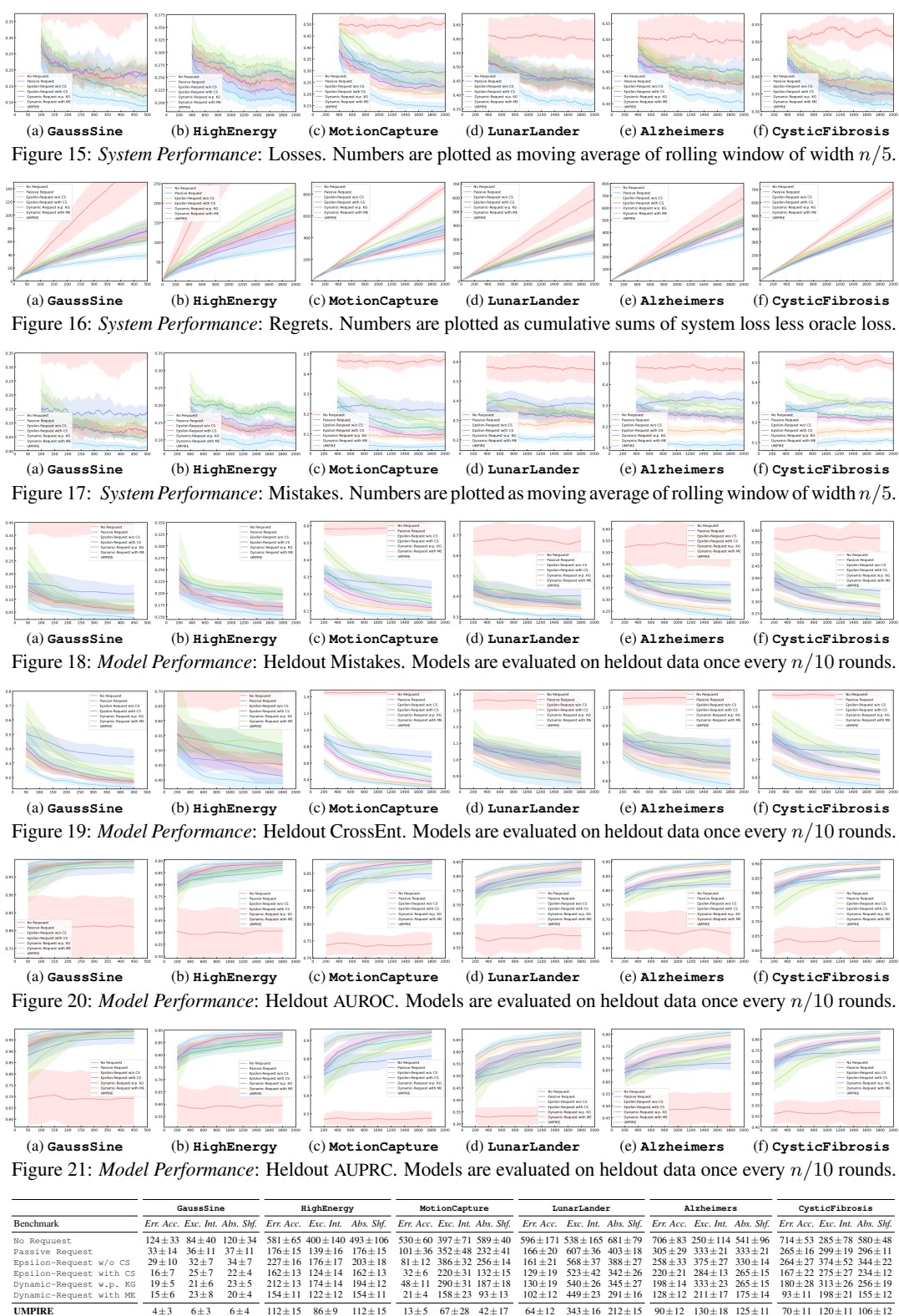

(a) **GaussSine**  (b) **HighEnergy**  (c) **MotionCapture**  (d) **LunarLander**  (e) **Alzheimers**  (f) **CysticFibrosis**

Figure 15: *System Performance*: Losses. Numbers are plotted as moving average of rolling window of width $n/5$.

(a) **GaussSine**  (b) **HighEnergy**  (c) **MotionCapture**  (d) **LunarLander**  (e) **Alzheimers**  (f) **CysticFibrosis**

Figure 16: *System Performance*: Regrets. Numbers are plotted as cumulative sums of system loss less oracle loss.

(a) **GaussSine**  (b) **HighEnergy**  (c) **MotionCapture**  (d) **LunarLander**  (e) **Alzheimers**  (f) **CysticFibrosis**

Figure 17: *System Performance*: Mistakes. Numbers are plotted as moving average of rolling window of width $n/5$.

(a) **GaussSine**  (b) **HighEnergy**  (c) **MotionCapture**  (d) **LunarLander**  (e) **Alzheimers**  (f) **CysticFibrosis**

Figure 18: *Model Performance*: Heldout Mistakes. Models are evaluated on heldout data once every $n/10$ rounds.

(a) **GaussSine**  (b) **HighEnergy**  (c) **MotionCapture**  (d) **LunarLander**  (e) **Alzheimers**  (f) **CysticFibrosis**

Figure 19: *Model Performance*: Heldout CrossEnt. Models are evaluated on heldout data once every $n/10$ rounds.

(a) **GaussSine**  (b) **HighEnergy**  (c) **MotionCapture**  (d) **LunarLander**  (e) **Alzheimers**  (f) **CysticFibrosis**

Figure 20: *Model Performance*: Heldout AUROC. Models are evaluated on heldout data once every $n/10$ rounds.

(a) **GaussSine**  (b) **HighEnergy**  (c) **MotionCapture**  (d) **LunarLander**  (e) **Alzheimers**  (f) **CysticFibrosis**

Figure 21: *Model Performance*: Heldout AUPRC. Models are evaluated on heldout data once every $n/10$ rounds.

| Benchmark | GaussSine | | | HighEnergy | | | MotionCapture | | | LunarLander | | | Alzheimers | | | CysticFibrosis | | |
|---|---|---|---|---|---|---|---|---|---|---|---|---|---|---|---|---|---|---|
| | *Err. Acc.* | *Exc. Int.* | *Abs. Shf.* | *Err. Acc.* | *Exc. Int.* | *Abs. Shf.* | *Err. Acc.* | *Exc. Int.* | *Abs. Shf.* | *Err. Acc.* | *Exc. Int.* | *Abs. Shf.* | *Err. Acc.* | *Exc. Int.* | *Abs. Shf.* | *Err. Acc.* | *Exc. Int.* | *Abs. Shf.* |
| No Request | 124±33 | 84±40 | 120±34 | 581±65 | 400±140 | 493±106 | 530±60 | 397±71 | 589±40 | 596±171 | 538±165 | 681±79 | 706±83 | 250±114 | 541±96 | 714±53 | 285±78 | 580±48 |
| Passive Request | 33±14 | 36±11 | 37±11 | 176±15 | 139±16 | 176±15 | 101±36 | 352±48 | 232±41 | 166±20 | 607±36 | 403±18 | 305±29 | 333±21 | 333±21 | 265±16 | 299±19 | 296±11 |
| Epsilon-Request w/o CS | 29±10 | 32±7 | 34±7 | 227±16 | 176±17 | 203±18 | 81±12 | 386±32 | 256±14 | 161±21 | 568±37 | 388±27 | 258±33 | 375±27 | 330±14 | 264±27 | 374±52 | 344±22 |
| Epsilon-Request with CS | 16±7 | 25±7 | 22±4 | 162±13 | 124±14 | 162±13 | 32±6 | 220±31 | 132±15 | 129±19 | 523±42 | 342±26 | 220±21 | 284±13 | 265±15 | 167±22 | 275±27 | 234±12 |
| Dynamic-Request w.p. KG | 19±5 | 21±6 | 23±5 | 212±13 | 174±14 | 194±12 | 48±11 | 290±31 | 187±18 | 130±19 | 540±26 | 345±27 | 198±14 | 333±23 | 265±15 | 180±28 | 313±26 | 256±19 |
| Dynamic-Request with ME | 15±6 | 23±8 | 20±4 | 154±11 | 122±12 | 154±11 | 21±4 | 158±23 | 93±13 | 102±12 | 449±23 | 291±16 | 128±12 | 211±17 | 175±14 | 93±11 | 198±21 | 155±12 |
| **UMPIRE** | 4±3 | 6±3 | 6±4 | 112±15 | 86±9 | 112±15 | 13±5 | 67±28 | 42±17 | 64±12 | 343±16 | 212±15 | 90±12 | 130±18 | 125±11 | 70±11 | 120±11 | 106±12 |

Table 5: *Mediator Performance*: Erroneous acceptance, excessive intervention, and abstention shortfall at $t = n$.

## A.3 Sensitivity Analysis

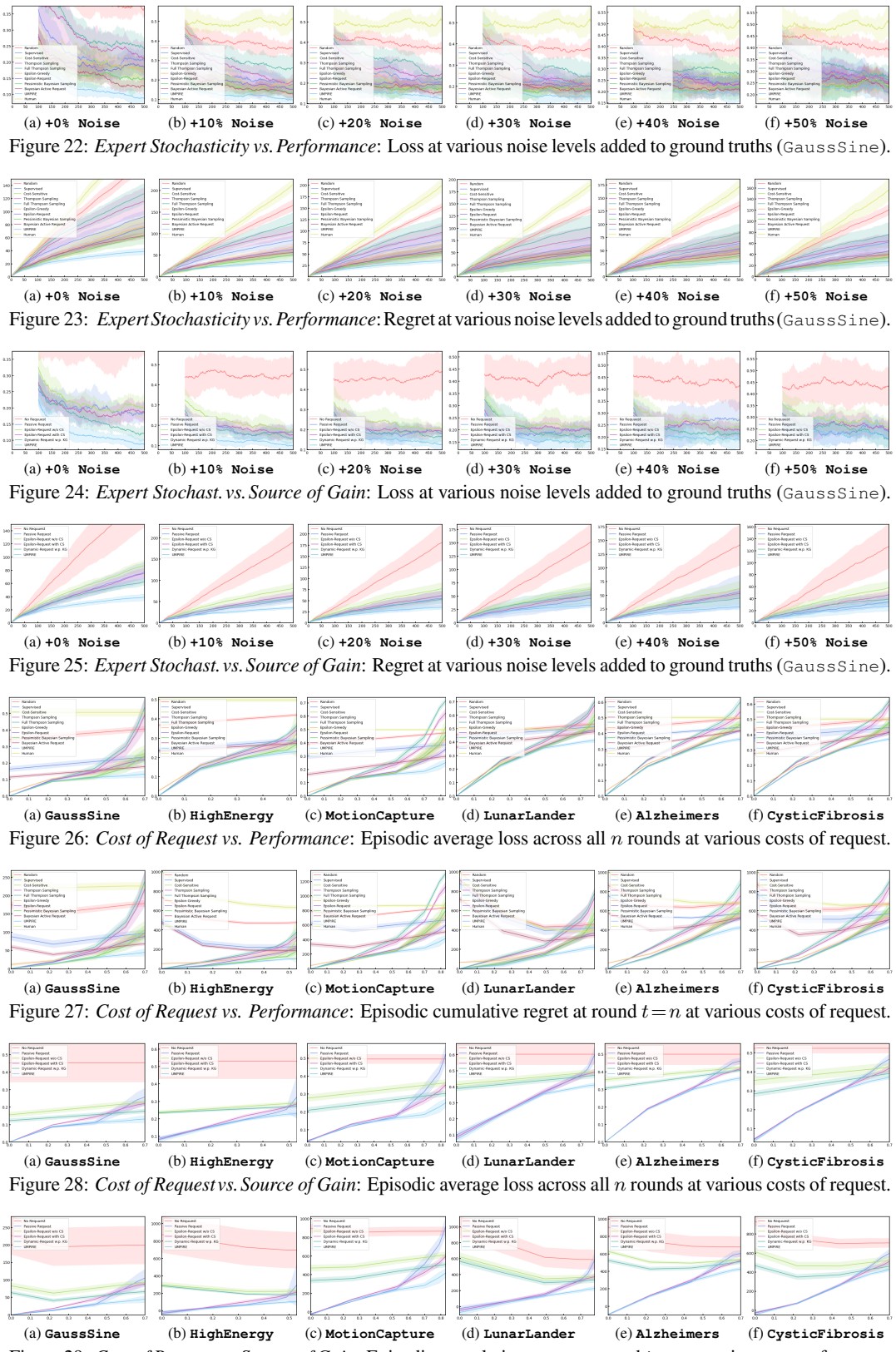

Figure 22: *Expert Stochasticity vs. Performance*: Loss at various noise levels added to ground truths (GaussSine).

(a) +0% Noise    (b) +10% Noise    (c) +20% Noise    (d) +30% Noise    (e) +40% Noise    (f) +50% Noise

Figure 23: *Expert Stochasticity vs. Performance*: Regret at various noise levels added to ground truths (GaussSine).

(a) +0% Noise    (b) +10% Noise    (c) +20% Noise    (d) +30% Noise    (e) +40% Noise    (f) +50% Noise

Figure 24: *Expert Stochast. vs. Source of Gain*: Loss at various noise levels added to ground truths (GaussSine).

(a) +0% Noise    (b) +10% Noise    (c) +20% Noise    (d) +30% Noise    (e) +40% Noise    (f) +50% Noise

Figure 25: *Expert Stochast. vs. Source of Gain*: Regret at various noise levels added to ground truths (GaussSine).

(a) GaussSine    (b) HighEnergy    (c) MotionCapture    (d) LunarLander    (e) Alzheimers    (f) CysticFibrosis

Figure 26: *Cost of Request vs. Performance*: Episodic average loss across all $n$ rounds at various costs of request.

(a) GaussSine    (b) HighEnergy    (c) MotionCapture    (d) LunarLander    (e) Alzheimers    (f) CysticFibrosis

Figure 27: *Cost of Request vs. Performance*: Episodic cumulative regret at round $t = n$ at various costs of request.

(a) GaussSine    (b) HighEnergy    (c) MotionCapture    (d) LunarLander    (e) Alzheimers    (f) CysticFibrosis

Figure 28: *Cost of Request vs. Source of Gain*: Episodic average loss across all $n$ rounds at various costs of request.

(a) GaussSine    (b) HighEnergy    (c) MotionCapture    (d) LunarLander    (e) Alzheimers    (f) CysticFibrosis

Figure 29: *Cost of Request vs. Source of Gain*: Episodic cumulative regret at round $t = n$ at various costs of request.

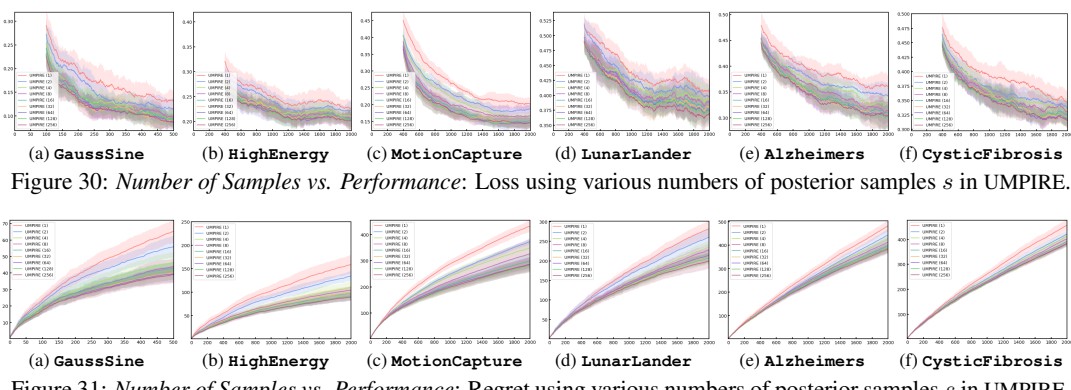

(a) `GaussSine`  (b) `HighEnergy`  (c) `MotionCapture`  (d) `LunarLander`  (e) `Alzheimers`  (f) `CysticFibrosis`

Figure 30: *Number of Samples vs. Performance*: Loss using various numbers of posterior samples $s$ in UMPIRE.

(a) `GaussSine`  (b) `HighEnergy`  (c) `MotionCapture`  (d) `LunarLander`  (e) `Alzheimers`  (f) `CysticFibrosis`

Figure 31: *Number of Samples vs. Performance*: Regret using various numbers of posterior samples $s$ in UMPIRE.

# B Experiment Details

## B.1 Dataset Details

All datasets used are publicly available. `GaussSine` is a synthetic environment that generates points by first constructing a smooth underlying latent function applied to two-dimensional iid. Gaussian input, and then setting decision boundaries by rounding the latent function to the nearest integer: $y = \text{round}(f(x_1, x_2))$ where $f(x_1, x_2) = \sin(0.15\pi u + x_1 + x_2) + 1$ with $u \sim \text{uniform}(0, 1)$; this is implemented exactly as given in the example in [61]. `HighEnergy` is a UCI dataset, and the task is to identify patterns caused by primary gammas (signal) from cosmic rays in the upper atmosphere (background), where the patterns are generated by simulated registration of high energy particles in an atmospheric Cherenkov gamma telescope [62]: There are $\approx 19000$ data points in total, from which $n = 2000$ are sampled for each randomly seeded run of experiments; the context space consists of 11 attributes, and the decision space is binary (signal vs. background). `MotionCapture` is a UCI dataset, and the task is to recognize different hand postures from data recorded by glove markers attached to users performing different movements using a Vicon motion capture camera system [63]: There are $\approx 78000$ data points in total, from which $n = 2000$ are sampled for each randomly seeded run of experiments; the context space consists of 38 attributes, and the decision space consists of 5 hand postures. In `LunarLander`, the task is to perform actions in the OpenAI gym [64] Atari environment "Lunar Lander": Given any game state, the expert action is defined as the action chosen by a PPO2 agent [65, 66] pre-trained on the environment using the true reward function. We generate $\approx 24000$ data points in total, from which $n = 2000$ are sampled for each randomly seeded run of experiments; the context space consists of 8 attributes, and the decision space consists of 4 rocket actions. The `Alzheimers` dataset records patients in the Alzheimer's Disease Neuroimaging Initiative study [67], including a range of demographic variables (age, education level, marital status, etc.), biomarkers (entorhinal, fusiform, hippocampus, etc.), and cognitive test scores (ADAS, CRD sum of boxes, mini mental state, etc.), and where the task is to perform early diagnosis of those patients as cognitively normal, mildly impaired, or at risk of dementia [19, 20]. There are $\approx 12000$ data points in total, from which $n = 2000$ are sampled for each randomly seeded run of experiments; the context space consists of 22 attributes, and the decision space consists of 3 diagnostic actions. The `CysticFibrosis` dataset records a cohort of patients enrolled in the UK Cystic Fibrosis registry [68], including a range of demographic variables (age, weight, smoking status, etc.), bacterial infections (burkholderia cepacia, pseudomonas aeruginosa, haemophilus influenza, etc.), and comorbidities (liver disease, hypertension, osteopenia, etc.), and where the task is to perform diagnosis of patients as to their GOLD grading in chronic obstructive pulmonary disease [69] (precisely, "severe" is up to 49% of predicted FEV1 value, "moderate" is between 50–79% of predicted FEV1 value, and "mild" otherwise). There are $\approx 31000$ data points in total, from which $n = 2000$ are sampled for each randomly seeded run of experiments; the context space consists of 34 attributes, and the decision space consists of 3 diagnostic actions.

## B.2 Benchmark Details

In the following, we note where algorithms previously studied for related fields in Table 1 are applicable to ODM or not. We implement the `Supervised` benchmark as picking $z \in \{0, 1\}$ based solely on $\hat{\pi}$'s output, and $z = 2$ w.p. $\epsilon$, which makes it resemble "supervised learning" in that incoming data

points for learning are not related to the algorithm's decisions whatsoever. The **Cost-Sensitive** benchmark is the greedy $(\hat{\pi}, \phi_*)$ from Section 3, which is a straightforward method for solving the "learning with rejection" problem [21–26], and does account for costs ($k_{\text{int}}$ and) $k_{\text{req}}$. However, any such approach has the obvious shortcoming that exploration is entirely passive, as noted in Section 3. Note that algorithms for "online learning with rejection" [27–30] cannot be applied to ODM since they require feedback that is always observed, whereas we work in a setting with abstentive feedback (moreover, existing algorithms are specialized to the binary case). In terms of active learning, note that the majority of related work is specialized to the binary setting in the interest of providing guarantees under specific assumptions; this is similar in environments without the rejection option [31–36] as well as with the rejection option [37–40]. However, a popular and practically applicable paradigm for active learning in the batch learning setting is the Bayesian active learning technique [60], which has since been extended to operate in a variety of other settings [75–77]. In this (pool-based) active learning setting, points are selected for query by ranking the expected reduction in entropy of the available points. Our **Bayesian Active Request** benchmark is implemented so as to most straightforwardly adapt this technique to the (stream-based) ODM problem by requesting w.p. $\sim$ expected reduction in entropy. Lastly, as noted in Section 2, the learning setting of [41] somewhat resembles ODM; however, their algorithm requires solving an intractable optimization problem over the version space, and is only applied to binary classification problems, which is not compatible with our more general requirement.

The class of algorithms most readily applicable—prima facie, at least—to ODM is stochastic contextual bandits. We implement the **Thompson Sampling** [46] benchmark straightforwardly: First, we draw a sample from the posterior $p(W|d)$, then we select $z$ greedily using the policy $(\hat{\pi}_W, \phi_*)$. There are two versions to consider: When $z = 1$ is chosen (i.e. when the model policy is asked to predict), we can either switch to using the marginal $p(Y|D, X) = \mathbb{E}_{W \sim p(\cdot|D)} p(Y|X, W)$ (which would be lower-variance), or still use the sample $\hat{\pi}_W$ (which would be canonical Thompson sampling); the latter is what we refer to as the **Full** version. Likewise, we implement the **Epsilon-Greedy** [47] benchmark straightforwardly: It acts exactly as the greedy mediator, but draws $z$ at random w.p. $\epsilon$. Given the discussion in Section 3.1, it should be clear that randomly exploring arms $z \in \{0, 1\}$ may be wasteful, therefore we also implement a **Request** version, which corresponds to the "$\epsilon$-request" described in Section 3.2: It acts exactly as the greedy mediator, but chooses $z = 2$ w.p. $\epsilon$. In addition to posterior sampling and epsilon-greedy policies, a third class of strategies in bandit settings is "optimism in the face of uncertainty". However, naively applying these (e.g. the UCB algorithm) would mean that arms $z \in \{0, 1\}$ are pulled *more* than in the greedy policy, and thus the request arm is pulled *less* (because model uncertainty is only involved in arms $z \in \{0, 1\}$, whereas there is no uncertainty when the request arm is pulled)—which is the exact opposite of what we want. Instead, we implement the **Pessimistic Bayesian Sampling** benchmark, which adapts OBS [44] to ODM by reversing the direction of optimism such that the tendency to request actually *increases*. Finally, to highlight the advantage of our proposed criterion, **Matched Decaying Request** is an artificially boosted benchmark that is similar to $\epsilon$-request—but where $\epsilon$ is a decay function that has the benefit of matching the effective request rate of UMPIRE: First, the behavior of UMPIRE is observed throughout its 10 runs of experiments for each environment. Second, we plot the curve containing the cumulative number of requests made by UMPIRE as a function of time, averaged across those 10 runs. Third, we search for a polynomial function $\epsilon_t$ of $t$ that—when used to define an $\epsilon_t$-request mediator—produces a request curve that best matches UMPIRE's request curve. (Thus we note that this benchmark is artificial in the sense that it has the benefit of hindsight knowledge).

## C  Proofs and Derivations

### C.1  Expected Improvement

Our proof technique is inspired by the strategy used to derive a deterministic generalization bound in [78]. However, there are several key differences: While they use a similar argument to derive a stopping criterion in the conventional active learning problem, we do so to derive a proxy for determining which mediator action to take in UMPIRE for the ODM problem. More importantly, their result gives a bound having observed both $x_t$ and $y_t$, whereas we work with only $x_t$ having been observed, with $y_t$ not yet observed. Furthermore, they explicitly compute KL divergences between parameter posteriors, whereas—with the expectation over $Y_t \sim p(\cdot|d_{t-1}, x_t)$ around the KL-term—we obtain an expression for mutual information instead, which allows us to expand it in the opposite direction to be amenable to computation, and does not require explicit KL divergences between parameter posteriors.

**Theorem 1** Let the model risk $\mathcal{R}$ be bounded as $[-b, b]$—for instance, by centering the loss $\ell_{01}$. Let $\mathbb{I}[W; Y_t | d_{t-1}, x_t]$ denote the mutual information between $W$ and $Y_t$ conditioned on $d_{t-1}$ and $x_t$, and let $W_0$ denote the principal branch of the product logarithm function. Then we have the following:

$$\bar{\mathcal{R}}(d_{t-1}) - \mathbb{E}_{Y_t \sim p(\cdot | d_{t-1}, x_t)}[\bar{\mathcal{R}}(D_t) | d_{t-1}, x_t, Z_t = 2] \leq 2b(e^{W_0\left(\frac{1}{e}(\mathbb{I}[W; Y_t | d_{t-1}, x_t] - 1)\right) + 1} - 1) \quad (6)$$

*Proof.* Let $f : \mathcal{W} \to [-b, b]$ and $c$ be a constant that satisfies $\mathbb{E}_{W \sim p(\cdot | d)} f(W)^2 \leq c$ for any $d \in \mathcal{D}$:

$$\mathbb{E}_{\substack{Y_t \sim p(\cdot | d_{t-1}, x_t) \\ W \sim p(\cdot | d_{t-1}, x_t, Y_t)}} f(W) - \mathbb{E}_{W \sim p(\cdot | d_{t-1})} f(W) \quad (8)$$

$$\leq \frac{1}{\lambda} \big( \mathbb{E}_{Y_t \sim p(\cdot | d_{t-1}, x_t)} \log \mathbb{E}_{W \sim p(\cdot | d_{t-1})} e^{\lambda f(W)} - \mathbb{E}_{W \sim p(\cdot | d_{t-1})} \lambda f(W)$$
$$+ \mathbb{E}_{Y_t \sim p(\cdot | d_{t-1}, x_t)} D_{\mathrm{KL}}\big(p(W | d_{t-1}, x_t, Y_t) \| p(W | d_{t-1})\big)\big) \quad (9)$$

$$\leq \frac{1}{\lambda}\big( (e^{2b\lambda} - 2b\lambda - 1) \mathbb{E}_{W \sim p(\cdot | d_{t-1})} \frac{f(W)^2}{4b^2}$$
$$+ \mathbb{E}_{Y_t \sim p(\cdot | d_{t-1}, x_t)} D_{\mathrm{KL}}\big(p(W | d_{t-1}, x_t, Y_t) \| p(W | d_{t-1})\big)\big) \quad (10)$$

$$\leq \frac{1}{\lambda}\big( \frac{c}{4b^2}(e^{2b\lambda} - 2b\lambda - 1) + \mathbb{I}[W; Y_t | d_{t-1}, x_t]\big) \quad (11)$$

where the first inequality uses Lemma 2, the second Lemma 3, and the third the fact that:

$$\mathbb{I}[W; Y_t | d_{t-1}, x_t] = \mathbb{E}_{Y_t \sim p(\cdot | d_{t-1}, x_t)} D_{\mathrm{KL}}(p(W | d_{t-1}, x_t, Y_t) \| p(W | d_{t-1})) \quad (12)$$

Minimizing Expression 11 with respect to $\lambda$ by setting the derivative to zero:

$$2be^{2b\lambda} - 2b = \frac{1}{\lambda}\big(e^{2b\lambda} - 2b\lambda - 1 + \frac{4b^2}{c}\mathbb{I}[W; Y_t | d_{t-1}, x_t]\big) \quad (13)$$

$$(2b\lambda - 1)e^{2b\lambda - 1} = \frac{4b^2}{ec}\big(\mathbb{I}[W; Y_t | d_{t-1}, x_t] - \frac{c}{4b^2}\big) \quad (14)$$

$$\lambda = \frac{1}{2b}\big(W_0\big(\frac{4b^2}{ec}\mathbb{I}[W; Y_t | d_{t-1}, x_t] - \frac{1}{e}\big) + 1\big) \quad (15)$$

Substituting this $\lambda$ back into Expression 11:

$$\frac{1}{\lambda}\big(e^{2b\lambda} - 2b\lambda - 1 + \mathbb{I}[W; Y_t | d_{t-1}, x_t]\big) \quad (16)$$

$$= \frac{2b}{W_0(h) + 1}\big(e^{W_0(h) + 1} - (W_0(h) + 1) + \mathbb{I}[W; Y_t | d_{t-1}, x_t] - 1\big) \quad (17)$$

$$= \frac{2b}{W_0(h) + 1}\big(e^{W_0(h) + 1} + W_0(h)e^{W_0(h) + 1} - (W_0(h) + 1)\big) \quad (18)$$

$$= 2b\big(e^{W_0(\frac{1}{e}\mathbb{I}[W; Y_t | d_{t-1}, x_t] - \frac{1}{e}) + 1} - 1\big) \quad (19)$$

where we define $h := \frac{1}{e}\mathbb{I}[W; Y_t | d_{t-1}, x_t] - \frac{1}{e}$ and let $c = 4b^2$. Now let $f := -\mathcal{R}$:

$$\bar{\mathcal{R}}(d_{t-1}) - \mathbb{E}_{Y_t \sim p(\cdot | d_{t-1}, x_t)}[\bar{\mathcal{R}}(D_t) | d_{t-1}, x_t, Z_t = 2]$$
$$= \mathbb{E}_{W \sim p(\cdot | d_{t-1})} \mathcal{R}(W) - \mathbb{E}_{\substack{Y_t \sim p(\cdot | d_{t-1}, x_t) \\ W \sim p(\cdot | d_{t-1}, x_t, Y_t)}} \mathcal{R}(W) \quad (20)$$

$$\leq g\big(\mathbb{I}[W; Y_t | d_{t-1}, x_t]\big)$$

where we define $g : v \mapsto g(v) = 2b(e^{W_0(\frac{1}{e}(v-1)) + 1} - 1)$, which concludes the proof. $\square$

**Lemma 2** For any $f : \mathcal{W} \to \mathbb{R}$:

$$\mathbb{E}_{W \sim p(\cdot | d_{t-1}, x_t, y_t)} f(W) \leq \log \mathbb{E}_{W \sim p(\cdot | d_{t-1})} e^{f(W)} + D_{\mathrm{KL}}\big(p(W | d_{t-1}, x_t, y_t) \| p(W | d_{t-1})\big) \quad (21)$$

*Proof.* Use any argument for Donsker-Varadhan's variational representation of the KL divergence, e.g.

$$D_{\mathrm{KL}}\big(p(W | d_{t-1}, x_t, y_t) \| p(W | d_{t-1})\big) \quad (22)$$

$$= \sup_{f' : \mathcal{W} \to \mathbb{R}} \big\{ \mathbb{E}_{W \sim p(\cdot | d_{t-1}, x_t, y_t)} f'(W) - \log \mathbb{E}_{W \sim p(\cdot | d_{t-1})} e^{f'(W)} \big\} \quad (23)$$

$$\geq \mathbb{E}_{W \sim p(\cdot | d_{t-1}, x_t, y_t)} f(W) - \log \mathbb{E}_{W \sim p(\cdot | d_{t-1})} e^{f(W)} \quad (24)$$

Or, to the same effect:

$$\log \mathbb{E}_{W \sim p(\cdot | d_{t-1})} e^{f(W)} \quad (25)$$

$$= \sup_{p' \in \Delta(\mathcal{W})} \big\{ \mathbb{E}_{W \sim p'} f(W) - D_{\mathrm{KL}}\big(p' \| p(W | d_{t-1})\big) \big\} \quad (26)$$

$$\geq \mathbb{E}_{W \sim p(\cdot | d_{t-1}, x_t, y_t)} f(W) - D_{\mathrm{KL}}\big(p(W | d_{t-1}, x_t, y_t) \| p(W | d_{t-1})\big) \quad (27)$$

**Lemma 3** For any $\lambda > 0$, $f: \mathcal{W} \to [-b, b]$:

$$\log \mathbb{E}_{W \sim p(\cdot|d_{t-1})} e^{\lambda f(W)} - \mathbb{E}_{W \sim p(\cdot|d_{t-1})} \lambda f(W) \leq (e^{2b\lambda} - 2b\lambda - 1) \mathbb{E}_{W \sim p(\cdot|d_{t-1})} \frac{f(W)^2}{4b^2} \quad (28)$$

*Proof.* Note that $\frac{1}{u^2}(e^u - u - 1)$ is a nondecreasing function of $u$, so:

$$e^{\lambda f(W)} - \lambda f(W) - 1 \leq (e^{2b\lambda} - 2b\lambda - 1) \frac{f(W)^2}{4b^2} \quad (29)$$

$$\mathbb{E}_{W \sim p(\cdot|d_{t-1})}[e^{\lambda f(W)} - \lambda f(W) - 1] \leq (e^{2b\lambda} - 2b\lambda - 1) \mathbb{E}_{W \sim p(\cdot|d_{t-1})} \frac{f(W)^2}{4b^2} \quad (30)$$

$$\log \mathbb{E}_{W \sim p(\cdot|d_{t-1})} e^{\lambda f(W)} - \mathbb{E}_{W \sim p(\cdot|d_{t-1})} \lambda f(W) \leq (e^{2b\lambda} - 2b\lambda - 1) \mathbb{E}_{W \sim p(\cdot|d_{t-1})} \frac{f(W)^2}{4b^2} \quad (31)$$

### C.2 Practical Implementation

We briefly elaborate here on the derivation for Equation 7. Let $\{w_{i,t}\}_{i=1}^s$ indicate the set of samples drawn from posterior $p(W|d_t)$. To compute the value of $\mathbb{I}[W; Y_t|d_{t-1}, x_t]$ at each round, we can write:

$$\mathbb{I}[W; Y_t|d_{t-1}, x_t] = \mathbb{E}_{Y_t \sim p(\cdot|d_{t-1}, x_t)} D_{\mathrm{KL}}(p(W|d_{t-1}, x_t, Y_t) \| p(W|d_{t-1})) \quad (32)$$

$$= \mathbb{H}[W|d_{t-1}] - \mathbb{E}_{Y_t \sim p(\cdot|d_{t-1}, x_t)} \mathbb{H}[W|d_{t-1}, x_t, Y_t] \quad (33)$$

which is the expected—over $Y_t \sim p(\cdot|d_{t-1}, x_t)$—reduction in entropy of $W$, i.e. how much the $Y_t$'s "inform" on $W$ given $x_t$. However, computing this term requires retraining on every possible $Y_t$ value:

$$H[p(W|d_{t-1})] - \sum_{y_t \in \mathcal{Y}} \mathbb{E}_{W \sim p(\cdot|d_{t-1})} p(y_t|x_t, W) H[\underbrace{p(W'|d_{t-1}, x_t, y_t)}_{\text{retrain}}] \quad (34)$$

where $H[p(W)] := -\frac{1}{s} \sum_{i=1}^s \log p(w_i)$. However, if we expand mutual information the opposite way:

$$\mathbb{I}[Y_t; W|d_{t-1}, x_t] = \mathbb{E}_{W \sim p(\cdot|d_{t-1})} D_{\mathrm{KL}}(p(Y_t|x_t, W) \| p(Y_t|d_{t-1}, x_{t-1})) \quad (35)$$

$$= \mathbb{H}[Y_t|d_{t-1}, x_t] - \mathbb{E}_{W \sim p(\cdot|d_{t-1})} \mathbb{H}[Y_t|x_t, W] \quad (36)$$

which is the expected—over $W \sim p(\cdot|d_{t-1})$—reduction in entropy of $Y_t|x_t$, i.e. how much the $W$'s "disagree" on $Y_t|x_t$ given $d_{t-1}$. Computing this term does not require retraining on every possible $Y_t$:

$$H[\tfrac{1}{s} \sum_{i=1}^s p(Y_t|x_t, w_{i,t-1})] - \tfrac{1}{s} \sum_{i=1}^s H[p(Y_t|x_t, w_{i,t-1})] \quad (37)$$

where $H[p(Y)] := -\sum_{y \in \mathcal{Y}} p(y) \log p(y)$. Therefore this is the implementation we use in Algorithm 1.

### C.3 Abstentive Feedback

In Section 3.1, we argued that abstentive feedback is the primary ingredient distinguishing ODM from bandits. To precisely highlight this, consider a more abstract formalism of a learner interacting with an environment. Let $\mathcal{X}$ be the space of contexts, $\mathcal{A}$ of actions, and $\mathcal{O}$ of outcomes, and let $\Sigma$ be some alphabet of feedback signals. Let $\mathbf{R} \in \mathbb{R}^{|\mathcal{A}| \times |\mathcal{O}|}$ denote the reward matrix, $\mathbf{F} \in \Sigma^{|\mathcal{A}| \times |\mathcal{O}|}$ the feedback matrix, and $\mathcal{H} := \Delta(\mathcal{O})^{\mathcal{X}}$ a class of functions mapping contexts to distributions of outcomes. Then:

**Definition 3** A *contextual partial monitoring* game [79, 80] between a learner and an environment is specified by $(\mathbf{R}, \mathbf{F}, \mathcal{H})$, which is given to the learner, and $h \in \mathcal{H}$, which is not. At the beginning of each round, a context $x_t \in \mathcal{X}$ is drawn exogenously, upon which the learner selects $a_t \in \mathcal{A}$ and the environment selects $o_t \in \mathcal{O}$. Then the learner receives the *unobservable* reward $\mathbf{R}_{a_t, o_t}$, as well as the *observable* feedback $\mathbf{F}_{a_t, o_t}$. The goal of the learner is to minimize the (unobserved) cumulative losses.

- Bandit Feedback: In this formalism, contextual bandit problems are characterized by the property that $\mathbf{F} = \mathbf{R}$: The feedback received for learning is identical to the loss incurred at every round. Therefore *some amount of learning always occurs*, regardless of which arm is ultimately pulled.

- Abstentive Feedback: In contrast, ODM is characterized by the fact that every row (i.e. action) of $\mathbf{F}$ contains the same (null) symbol for all columns (i.e. outcomes), except the row corresponding to requesting the expert decision. Writing out the matrices for ODM with mediator $(\hat{\pi}, \phi)$ as the learner:

$$\mathbf{R} = \begin{bmatrix} -\ell(y^{(0)}, \delta(Y - \tilde{y}_t)) & \cdots & -\ell(y^{(m-1)}, \delta(Y - \tilde{y}_t)) \\ -k_{\mathrm{int}} & \cdots & -\ell(y^{(m-1)}, \delta(Y - y^{(0)})) - k_{\mathrm{int}} \\ -\ell(y^{(0)}, \delta(Y - y^{(1)})) - k_{\mathrm{int}} & \cdots & -\ell(y^{(m-1)}, \delta(Y - y^{(1)})) - k_{\mathrm{int}} \\ \vdots & \cdots & \vdots \\ -\ell(y^{(0)}, \delta(Y - y^{(m-2)})) - k_{\mathrm{int}} & \cdots & -\ell(y^{(m-1)}, \delta(Y - y^{(m-2)})) - k_{\mathrm{int}} \\ -\ell(y^{(0)}, \delta(Y - y^{(m-1)})) - k_{\mathrm{int}} & \cdots & -k_{\mathrm{int}} \\ -k_{\mathrm{req}} & \cdots & -k_{\mathrm{req}} \end{bmatrix} \quad \mathbf{F} = \begin{bmatrix} \bot & \cdots & \bot \\ \bot & \cdots & \bot \\ \bot & \cdots & \bot \\ \vdots & \cdots & \vdots \\ \bot & \cdots & \bot \\ \bot & \cdots & \bot \\ y^{(0)} & \cdots & y^{(m-1)} \end{bmatrix} \quad (38)$$

where $y^{(i)}$ denotes the $i^{\text{th}}$ element in $\mathcal{Y}$ (containing $m$ elements), indexed from zero upwards. (The first row in each matrix corresponds to accepting the human decision, the last row requesting from the expert, and intermediate rows intervening with a model output; the columns correspond to the $m$ possible values that the oracle $Y$ can take.[1]) Thus *no learning occurs for all except one action*—which is why the role of "exploration" is very different in that most arms pulled for the sake of exploration yield no learning. Therefore we would not expect naively applying bandit algorithms in ODM to perform well.

Note that there does exist a rich literature on variations on the bandit problem—e.g. combinatorial bandits, bandits with side information, etc.—in which the feedback received at each round is *more* informative than the loss incurred. However, that only means the learner has strictly more information to work with. The point here is that contextual bandits never provide feedback with *less* information than the loss, which is the case for ODM. Also, note that while Definition 3 is general, only some special cases have been studied, such as apple tasting with a logistic model [55], locally-observable games with linear/logistic models [79], or where both rewards and observations are linear [80]. Future work may consider analyzing ODM under more general contextual partial monitoring settings. Lastly, note that partial monitoring is not to be confused with partially-observable bandits (see e.g. [81, 82]): The latter deals with partially-observed *contexts*, while the former deals with partially-observed *feedback*.

# D Further Related Work

Section 2.2 described the context and challenges of ODM, paying particular attention to the three primary strands of related problem settings: *learning with rejection* [21–30], *stream-based active learning* [31–36], *stochastic contextual bandits* [42–48], as well as some variations thereof [37–41, 49–58]. For completeness, we now describe some additional domains of—more tangentially—related work.

There is a wide-ranging field of research dedicated to studying various aspects of "assistive learning" and "cooperative learning", where the overarching goal is to use machines to help humans achieve their goals [83, 84]. For instance, *active reward learning* seeks to infer a human's goals, preferences, or reward functions by observing their behavior and choosing particular questions to ask the human for targeted feedback [85–88]. A typical solution consists of a question policy and a policy decision function to maximize the expected reward. On a different note, *assistance games* model humans as part of the environment with some latent goal, and the agent's goal is to balance between actions that learn about this goal and actions that achieve the learned goal [83, 89–91]. This leads to a two-agent POMDP with the goal of finding pairs of strategies for humans and machines that maximize expected reward, with human actions and machine actions observable to both. In either case, while the high-level idea is similar to ours—that is, to seek some notion of "principal-agent alignment" [92] between humans and machines—their formalisms, objectives, and solution strategies are entirely distinct from ODM.

In the supervised learning setting, *active label correction* deals with interactive methods for cleaning an established training dataset of possibly-mislabeled examples by using the assistance of a domain expert [93–95]. Since cleaned labels are only obtainable at a cost, the goal is to identify training patterns for which knowing the true labels best improves the learning algorithm's performance. For instance, [94] leverages the assumption of class-conditional noise, and [95] takes advantage of the assumption that noise is concentrated near decision boundaries. In the incremental learning setting, *skeptical learning* deals with learning from a stream of possibly-mislabeled examples, where the algorithm has the opportunity to ask the human to double-check their annotations before learning from each incoming data point [96–98]. For instance, [96] uses an estimate of confidence about both the model and the user in order to decide whether or not an example is worth double-checking, and [97] uses Gaussian processes to supply explicit uncertainty estimates regarding each incoming sample. In either case, a pitfall is that noisy labels may sometimes be admitted and erroneously learned from, which means that incorrect data accumulates over time with high probability. The idea of label correction has also been applied to *active learning from weak labelers*, either given the ability to query from a pool of samples [99], or the ability to query anywhere in the input space [100]. In all these fields, while the high-level idea of interacting with strong (oracle) and weak (human) agents bears some resemblance to our setting, their formalisms, objectives, and solution strategies are entirely distinct from ODM.

---

[1] Here, the space $\mathcal{A}$ contains the $m+2$ "arms" (i.e. one *accept* arm, one *request* arm, and one *intervene* arm for each of the $m$ underlying actions), the space $\mathcal{O}=\mathcal{Y}$ contains the $m$ possible expert ground-truths, and the alphabet $\Sigma=\mathcal{Y}\cup\{\perp\}$ contains the $m$ possible labels, as well as including the null symbol to denote the lack of feedback.

# E Further Discussions and Sensitivities

## E.1 Why keep the imperfect human in the loop?

Our choice to keep the imperfect human in the loop is motivated by real-world ethical considerations: In many high-stakes settings, concrete attribution of *responsibility* is an absolute requirement. For instance, in most healthcare systems, the responsibility for a cancer diagnosis must be traceable to the *person* (not a *machine*) who actually ordered it—and is therefore legally/ethically/financially accountable for it (see e.g. [13, 15]).

Our formulation of the ODM problem reflects this consideration. In our framework, an imperfect human first attempts to make a decision, which is observed. Then, the *mediator* decides which of the following happens: (1) "Accept": The imperfect human's decision goes through. So the *imperfect human* is responsible for that decision. (2) "Intervene": The mediator proposes an alternative to the imperfect human. This immediately incurs a cost ($k_{int}$), representing the fact that the imperfect human is now asked to spend time/resources in reconsidering/altering their original decision in light of the proposal. But the *imperfect human* is still responsible for the decision, regardless of whether or not they comply with the proposal.[2] (3) "Request": The expert is invited to make the decision instead. So the *expert* is now responsible for that decision.

If we "remove" the imperfect human from the loop, then all decisions would be made autonomously by the machine unless the expert is queried—which is strongly at odds with societal notions of ethical responsibility/accountability for high-stakes decisions. (To be clear, the "intervene" option is not autonomously "overriding" the imperfect human's decision: It is simply proposing an alternative!)

## E.2 Can the model interact only with the expert?

Many existing works operate in a framework where the *machine* itself is directly allowed to make decisions autonomously, and selectively query the expert (e.g. when it is uncertain). Given the above discussion, we would expect that this setting is generally applicable to non-high-stakes decisions (i.e. where notions of responsibility are not legally/ethically/financially required to be tied to a *person*).

That being said, the ODM framework is simply a *generalization* of this. Consider setting $k_{int} = 0$ in ODM: This effectively recovers the simpler setting where a machine interacts with an expert (i.e. the imperfect human would indeed become "redundant"). Importantly, however, in general the human is *not* "redundant" whenever $k_{int} > 0$, as long as the imperfect human has non-zero probability of being correct. Precisely, the mediator policy needs to decide if the human is likely already correct, so it can choose to "accept" (instead of "intervene") in order to avoid incurring the cost $k_{int}$.

Finally, it is worth reiterating that the ODM problem is also distinguished by the fact that feedback is *abstentive*. Whereas, existing works in machine-expert interaction (including [24] and [101]) operate in settings where feedback is *not* abstentive. That is to say, even if we set $k_{int} = 0$, the ODM setting is still fundamentally more challenging to tackle, as discussed throughout the manuscript.

**Preface to further results in Appendices E.3–E.6**

In our experiments, the imperfect human's decisions are simulated with $\alpha = 0.5$, although the specific frequency of human mistakes does not affect the basic structure/hardness of the problem. Our motivation is simply to simulate decisions that stochastically deviate from the expert's with some probability between zero and one. Appendices E.3–E.4 perform a complete re-run of our main experiments under the same conditions as before—but now setting $\alpha = 0.9$. For completeness, we can additionally ask how performance varies for different levels of $\alpha$, and if $k_{int} = 0.0$ instead of $k_{int} = 0.1$. Appendices E.5–E.6 perform experiments for the cartesian product of settings $\alpha \in \{0.5, 0.7, 0.9, 1.0\}$ and $k_{int} \in \{0.0, 0.1\}$, using the GaussSine environment. Across all sensitivities, it is easy to verify that UMPIRE still consistently accumulates lower regret (our primary metric of interest), as well as outperforming comparators with respect to to the rest of the performance measures.

---

[2] Whether or not they comply with the proposal is beyond the scope of our work: There may be a variety of reasons why they do/don't comply downstream. Importantly, however, from the perspective of the *mediator*, it has already done its job—and the correctness of its proposed alternative can be evaluated (and on the basis of which we can define concrete notions of model regret and system regret, which is what we do).

# E.3 Sensitivity on Human Error (Performance)

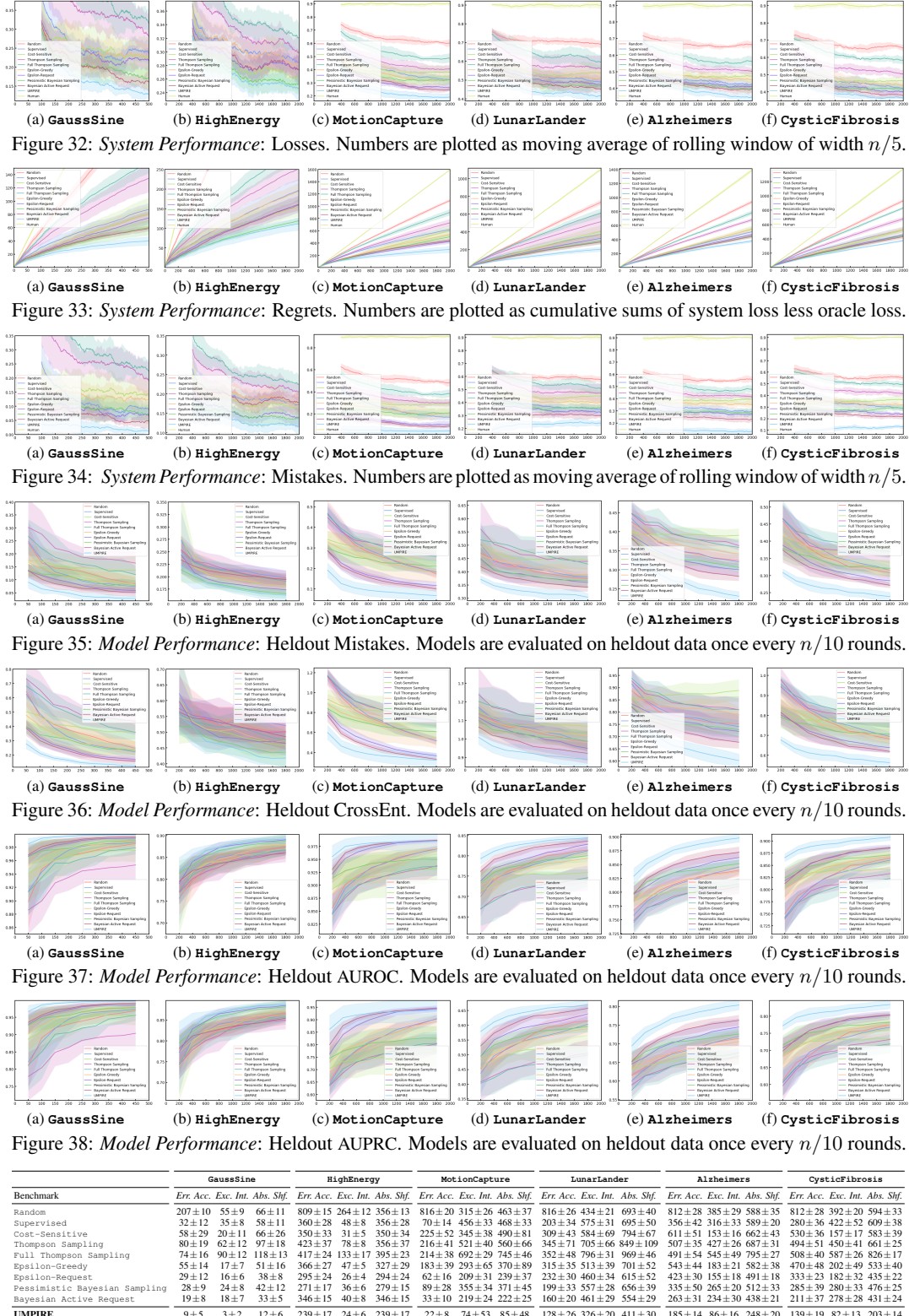

Figure 32: *System Performance*: Losses. Numbers are plotted as moving average of rolling window of width $n/5$.

Figure 33: *System Performance*: Regrets. Numbers are plotted as cumulative sums of system loss less oracle loss.

Figure 34: *System Performance*: Mistakes. Numbers are plotted as moving average of rolling window of width $n/5$.

Figure 35: *Model Performance*: Heldout Mistakes. Models are evaluated on heldout data once every $n/10$ rounds.

Figure 36: *Model Performance*: Heldout CrossEnt. Models are evaluated on heldout data once every $n/10$ rounds.

Figure 37: *Model Performance*: Heldout AUROC. Models are evaluated on heldout data once every $n/10$ rounds.

Figure 38: *Model Performance*: Heldout AUPRC. Models are evaluated on heldout data once every $n/10$ rounds.

| Benchmark | GaussSine Err. Acc. | Exc. Int. | Abs. Shf. | HighEnergy Err. Acc. | Exc. Int. | Abs. Shf. | MotionCapture Err. Acc. | Exc. Int. | Abs. Shf. | LunarLander Err. Acc. | Exc. Int. | Abs. Shf. | Alzheimers Err. Acc. | Exc. Int. | Abs. Shf. | CysticFibrosis Err. Acc. | Exc. Int. | Abs. Shf. |
|---|---|---|---|---|---|---|---|---|---|---|---|---|---|---|---|---|---|---|
| Random | 207±10 | 55±9 | 66±11 | 809±15 | 264±12 | 356±13 | 816±20 | 315±26 | 463±37 | 816±26 | 434±21 | 693±40 | 812±28 | 385±29 | 588±35 | 812±28 | 392±20 | 594±33 |
| Supervised | 32±12 | 35±8 | 58±11 | 360±28 | 48±8 | 356±28 | 70±14 | 456±33 | 468±33 | 203±34 | 575±31 | 695±50 | 356±42 | 316±33 | 589±20 | 280±36 | 422±52 | 609±38 |
| Cost-Sensitive | 58±29 | 20±11 | 66±26 | 350±33 | 31±5 | 350±34 | 225±52 | 345±38 | 490±81 | 309±43 | 584±69 | 794±67 | 611±51 | 153±16 | 662±43 | 530±36 | 157±17 | 583±39 |
| Thompson Sampling | 80±19 | 62±12 | 97±18 | 423±37 | 78±8 | 356±37 | 216±41 | 521±40 | 560±66 | 345±71 | 705±66 | 849±109 | 507±35 | 427±26 | 687±31 | 494±51 | 450±41 | 661±25 |
| Full Thompson Sampling | 74±16 | 90±12 | 118±13 | 417±24 | 133±17 | 395±23 | 214±38 | 692±29 | 745±46 | 352±48 | 796±31 | 969±46 | 491±54 | 545±49 | 795±27 | 508±40 | 587±26 | 826±17 |
| Epsilon-Greedy | 55±14 | 17±7 | 51±16 | 366±27 | 47±5 | 327±29 | 183±39 | 293±65 | 370±89 | 315±35 | 513±39 | 701±52 | 543±44 | 183±21 | 582±38 | 470±48 | 202±49 | 533±40 |
| Epsilon-Request | 29±12 | 16±6 | 38±8 | 295±24 | 26±4 | 294±24 | 62±16 | 209±31 | 239±37 | 232±30 | 460±34 | 615±52 | 423±30 | 155±18 | 491±18 | 333±23 | 182±32 | 435±22 |
| Pessimistic Bayesian Sampling | 28±9 | 24±8 | 42±12 | 271±17 | 36±6 | 279±15 | 89±28 | 355±34 | 371±45 | 199±33 | 557±28 | 656±39 | 335±50 | 265±20 | 512±33 | 285±39 | 280±33 | 476±25 |
| Bayesian Active Request | 19±8 | 18±7 | 33±5 | 346±15 | 40±8 | 346±15 | 33±10 | 219±24 | 222±25 | 160±20 | 461±29 | 554±29 | 263±31 | 234±30 | 438±21 | 211±37 | 278±28 | 431±24 |
| **UMPIRE** | 9±5 | 3±2 | 12±6 | 239±17 | 24±6 | 239±17 | 22±8 | 74±53 | 85±48 | 128±26 | 326±20 | 411±30 | 185±14 | 86±16 | 248±20 | 139±19 | 82±13 | 203±14 |

Table 6: *Mediator Performance*: Erroneous acceptance, excessive intervention, and abstention shortfall at $t = n$.

## E.4 Sensitivity on Human Error (Source of Gain)

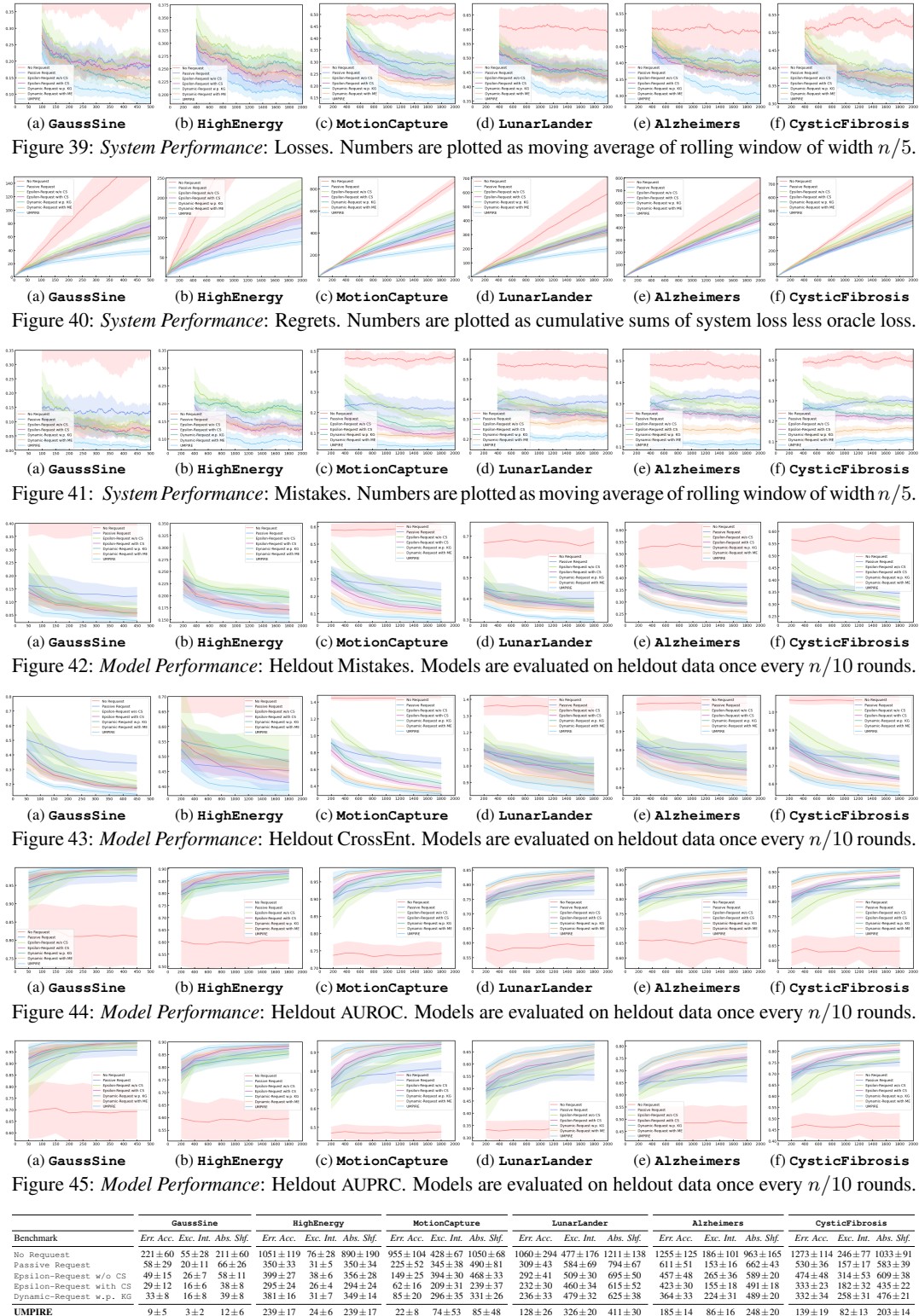

Figure 39: *System Performance*: Losses. Numbers are plotted as moving average of rolling window of width $n/5$.

Figure 40: *System Performance*: Regrets. Numbers are plotted as cumulative sums of system loss less oracle loss.

Figure 41: *System Performance*: Mistakes. Numbers are plotted as moving average of rolling window of width $n/5$.

Figure 42: *Model Performance*: Heldout Mistakes. Models are evaluated on heldout data once every $n/10$ rounds.

Figure 43: *Model Performance*: Heldout CrossEnt. Models are evaluated on heldout data once every $n/10$ rounds.

Figure 44: *Model Performance*: Heldout AUROC. Models are evaluated on heldout data once every $n/10$ rounds.

Figure 45: *Model Performance*: Heldout AUPRC. Models are evaluated on heldout data once every $n/10$ rounds.

| | GaussSine | | | HighEnergy | | | MotionCapture | | | LunarLander | | | Alzheimers | | | CysticFibrosis | | |
|---|---|---|---|---|---|---|---|---|---|---|---|---|---|---|---|---|---|---|
| Benchmark | *Err. Acc.* | *Exc. Int.* | *Abs. Shf.* | *Err. Acc.* | *Exc. Int.* | *Abs. Shf.* | *Err. Acc.* | *Exc. Int.* | *Abs. Shf.* | *Err. Acc.* | *Exc. Int.* | *Abs. Shf.* | *Err. Acc.* | *Exc. Int.* | *Abs. Shf.* | *Err. Acc.* | *Exc. Int.* | *Abs. Shf.* |
| No Request | 221±60 | 55±28 | 211±60 | 1051±119 | 76±28 | 890±190 | 955±104 | 428±67 | 1050±68 | 1060±294 | 477±176 | 1211±138 | 1255±125 | 186±101 | 963±165 | 1273±114 | 246±77 | 1033±91 |
| Passive Request | 58±29 | 20±11 | 66±26 | 350±33 | 31±5 | 350±34 | 225±52 | 345±38 | 490±81 | 309±43 | 584±69 | 794±67 | 611±51 | 153±16 | 662±43 | 530±36 | 157±17 | 583±39 |
| Epsilon-Request w/o CS | 49±15 | 26±7 | 58±11 | 399±27 | 38±6 | 356±28 | 149±25 | 394±30 | 468±33 | 292±41 | 509±30 | 695±50 | 457±48 | 265±36 | 589±20 | 474±48 | 314±53 | 609±38 |
| Epsilon-Request with CS | 29±12 | 16±6 | 38±8 | 295±24 | 26±4 | 294±24 | 62±16 | 209±31 | 239±37 | 232±30 | 460±34 | 615±52 | 423±30 | 155±18 | 491±18 | 333±23 | 182±32 | 435±22 |
| Dynamic-Request w.p. KG | 33±8 | 16±8 | 39±8 | 381±16 | 31±7 | 349±14 | 85±20 | 296±35 | 331±26 | 236±33 | 479±32 | 625±38 | 364±33 | 224±31 | 489±20 | 332±34 | 258±31 | 476±21 |
| **UMPIRE** | 9±5 | 3±2 | 12±6 | 239±17 | 24±6 | 239±17 | 22±8 | 74±53 | 85±48 | 128±26 | 326±20 | 411±30 | 185±14 | 86±16 | 248±20 | 139±19 | 82±13 | 203±14 |

Table 7: *Mediator Performance*: Erroneous acceptance, excessive intervention, and abstention shortfall at $t = n$.

## E.5 Complete Sensitivities on $\alpha$ and $k_{\text{int}}$ (Performance)

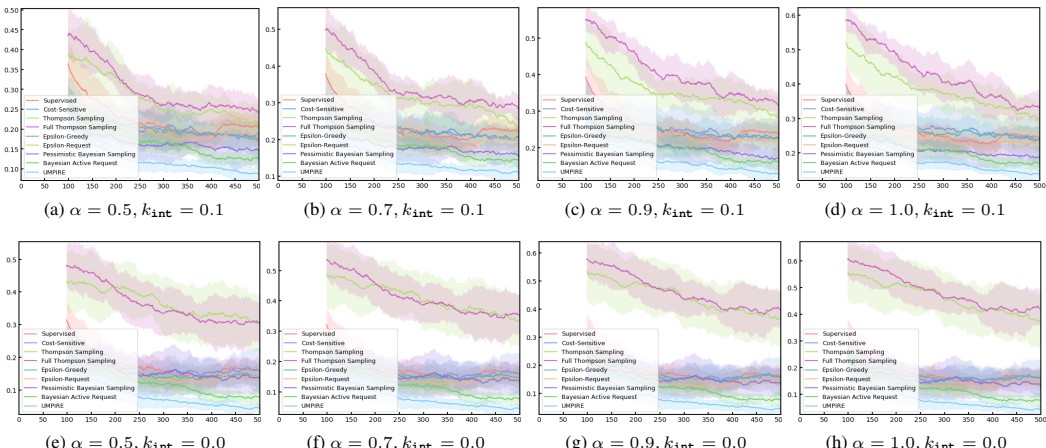

Figure 46: *System Performance*: Losses. Numbers are plotted as moving average of rolling window of width $n/5$.

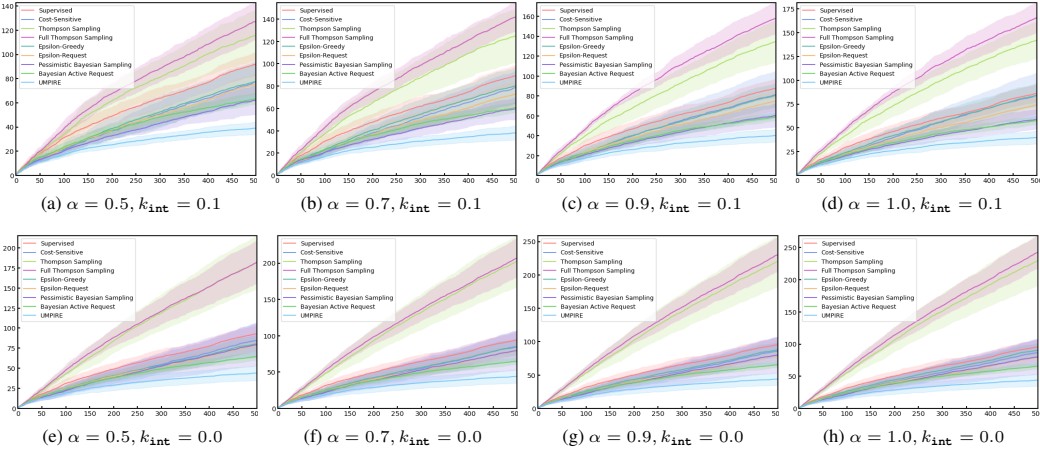

Figure 47: *System Performance*: Regrets. Numbers are plotted as cumulative sums of system loss less oracle loss.

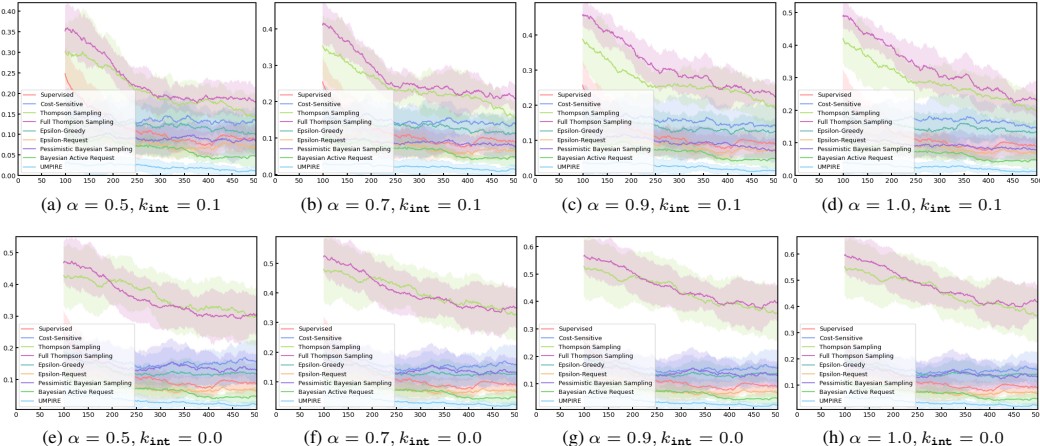

Figure 48: *System Performance*: Mistakes. Numbers are plotted as moving average of rolling window of width $n/5$.

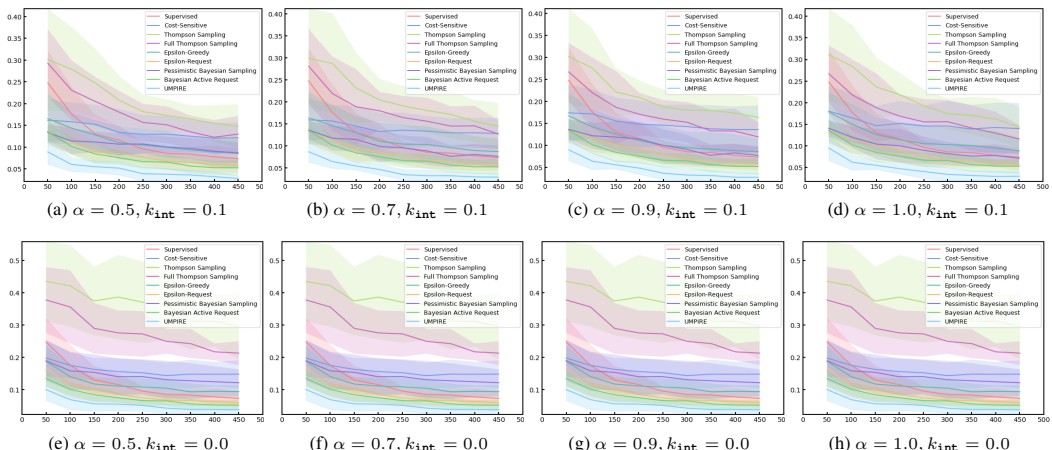

Figure 49: *Model Performance*: Heldout Mistakes. Models are evaluated on heldout data once every $n/10$ rounds.

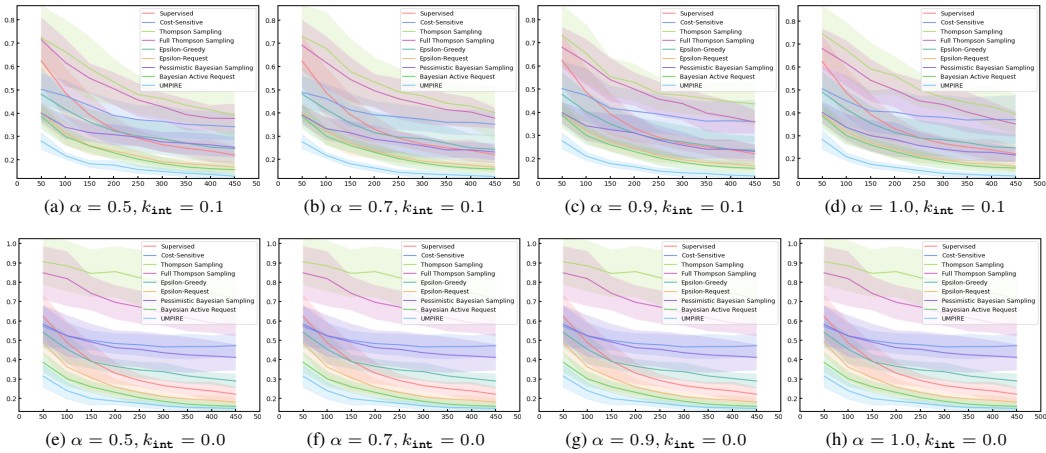

Figure 50: *Model Performance*: Heldout CrossEnt. Models are evaluated on heldout data once every $n/10$ rounds.

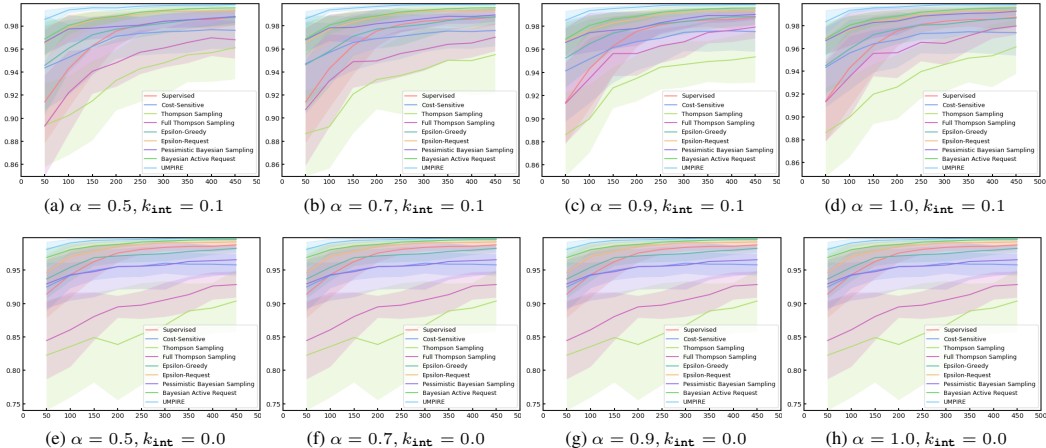

Figure 51: *Model Performance*: Heldout AUROC. Models are evaluated on heldout data once every $n/10$ rounds.

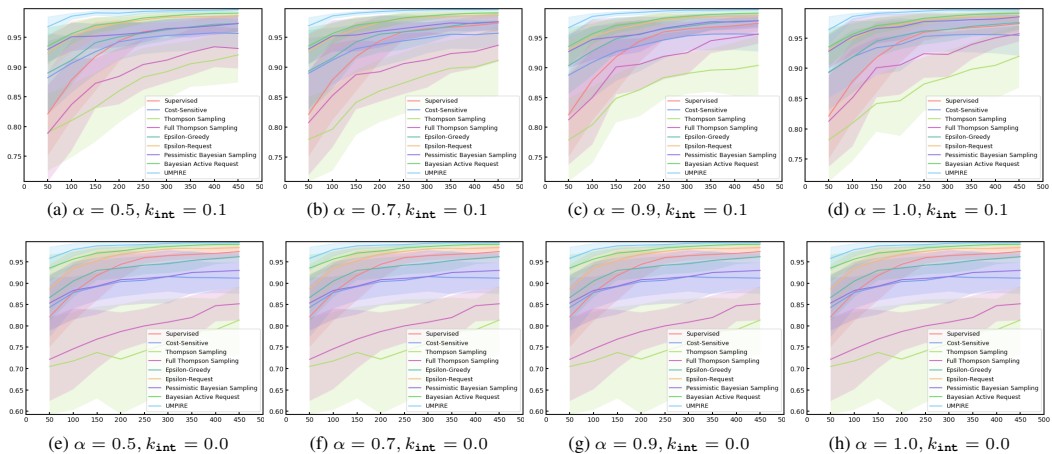

Figure 52: *Model Performance*: Heldout AUPRC. Models are evaluated on heldout data once every $n/10$ rounds.

## E.6 Complete Sensitivities on $\alpha$ and $k_{\text{int}}$ (Source of Gain)

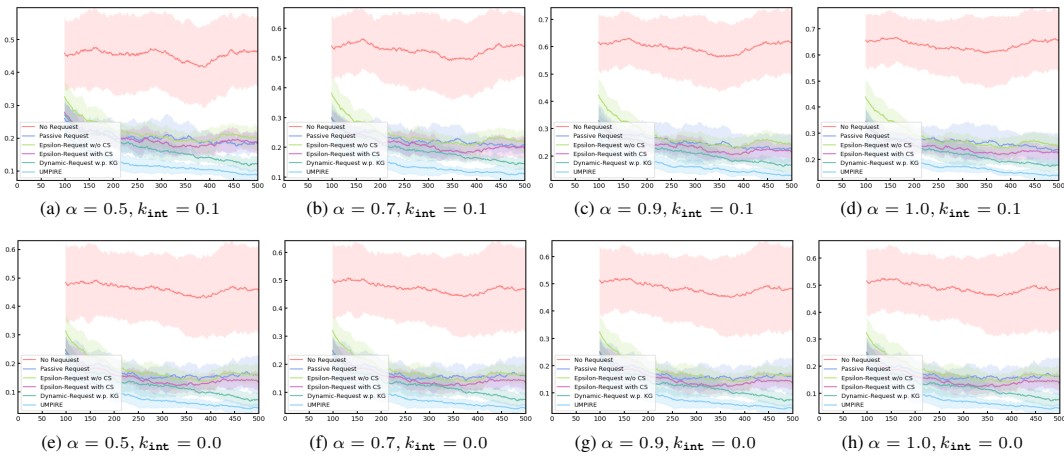

Figure 53: *System Performance*: Losses. Numbers are plotted as moving average of rolling window of width $n/5$.

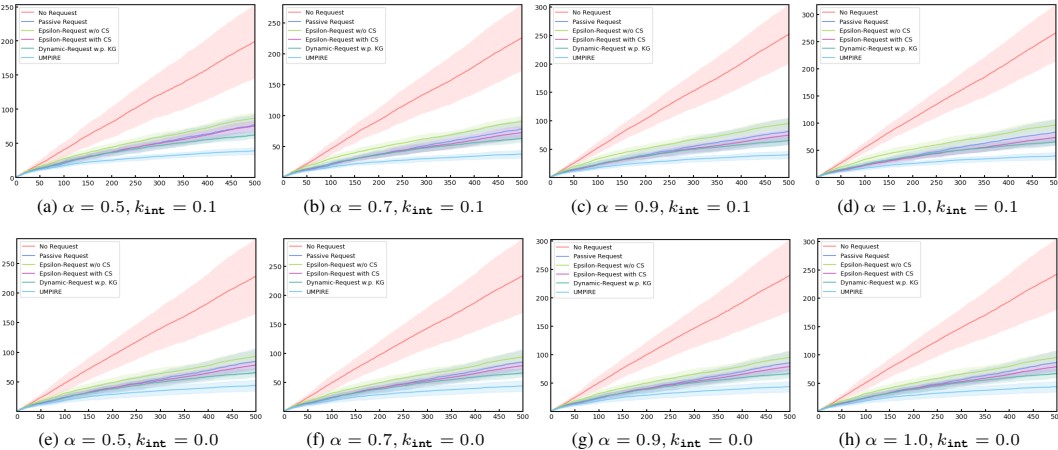

Figure 54: *System Performance*: Regrets. Numbers are plotted as cumulative sums of system loss less oracle loss.

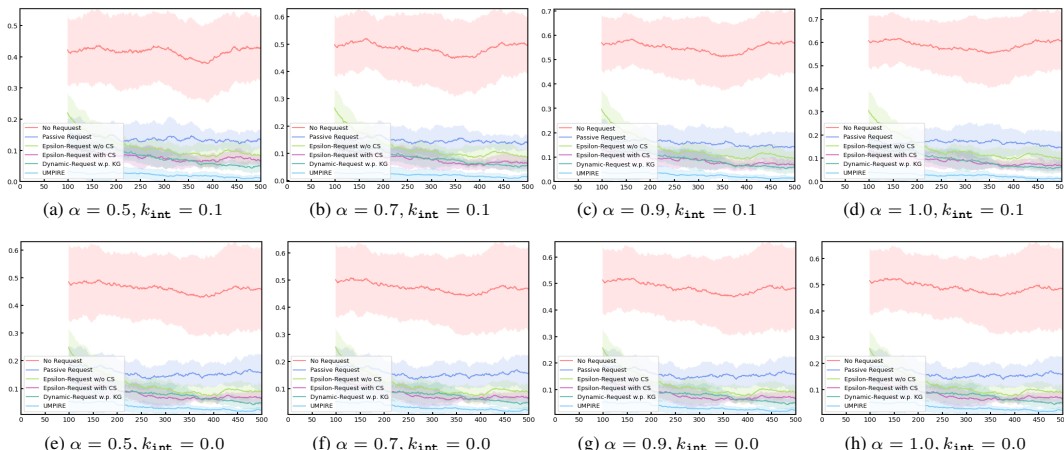

Figure 55: *System Performance*: Mistakes. Numbers are plotted as moving average of rolling window of width $n/5$.

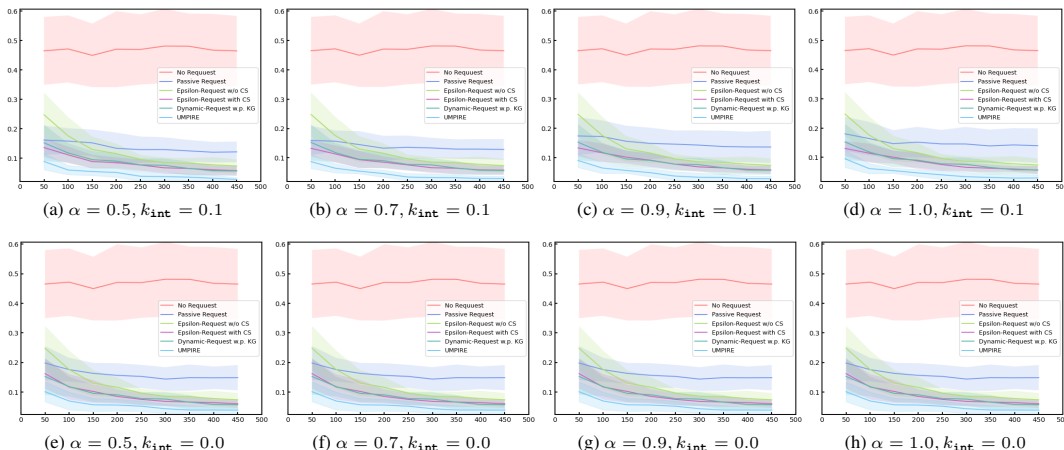

Figure 56: *Model Performance*: Heldout Mistakes. Models are evaluated on heldout data once every $n/10$ rounds.

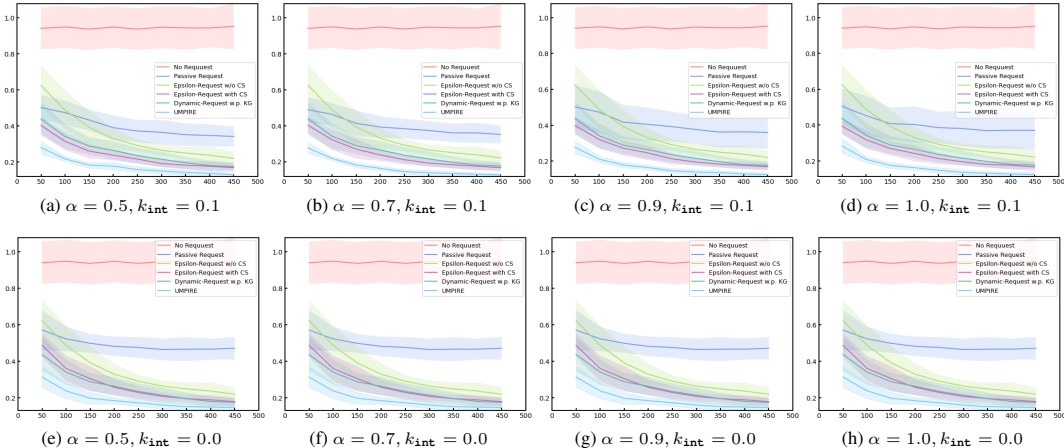

Figure 57: *Model Performance*: Heldout CrossEnt. Models are evaluated on heldout data once every $n/10$ rounds.

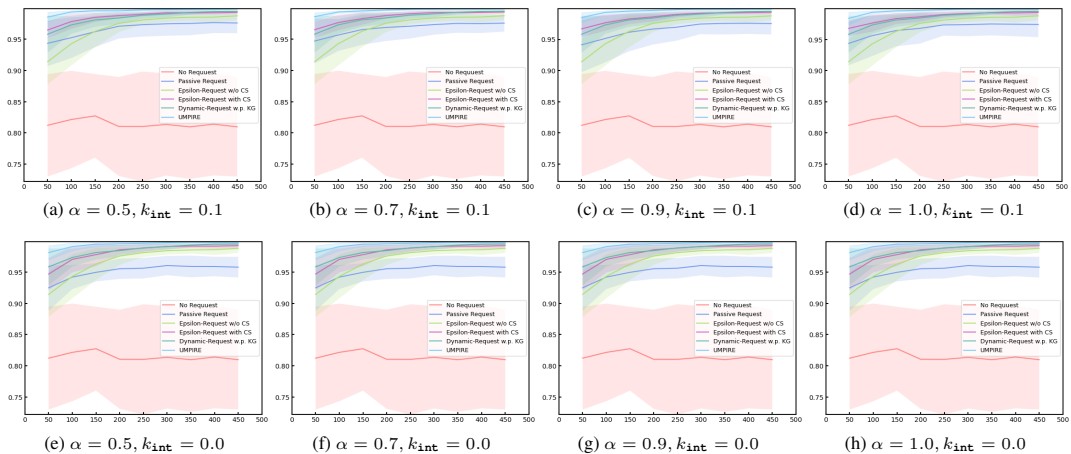

Figure 58: *Model Performance*: Heldout AUROC. Models are evaluated on heldout data once every $n/10$ rounds.

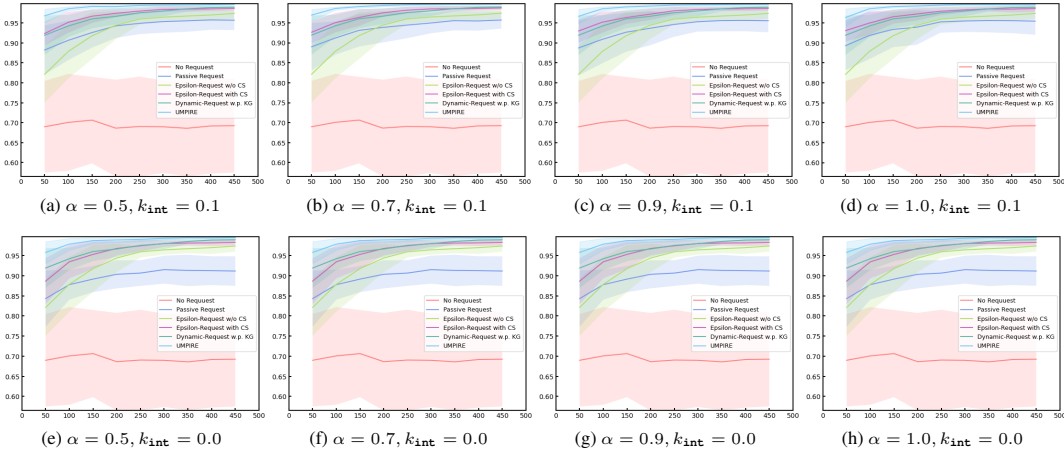

Figure 59: *Model Performance*: Heldout AUPRC. Models are evaluated on heldout data once every $n/10$ rounds.

### E.7 How Mediation Evolves over Time

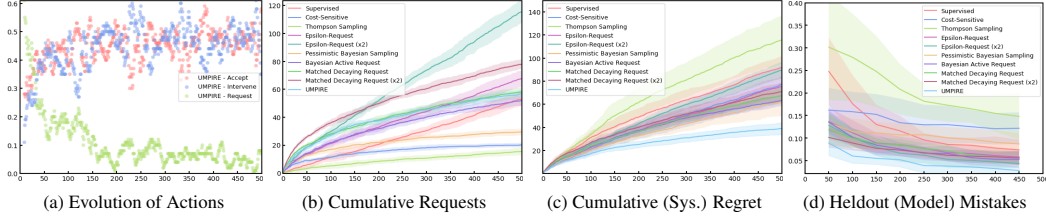

Figure 60: Visualization of Evolution of Actions, Cumulative Requests, and System/Model Performances

Intuitively, we would expect a good mediator policy to request more in the beginning, when the model is more likely to learn from expert actions (and more likely to make mistakes otherwise). Figure 60(a) shows how $Z$ evolves over time by visualizing the relative frequencies of each mediator action (i.e. "accept", "intervene", and "request") that UMPIRE issues, using GaussSine as environment. Numbers are plotted as moving averages of windows of length 10 (steps). Moreover, we can also ask how the pattern of "request" actions compares across different methods. Figure 60(b) plots the cumulative progression of request actions for UMPIRE, as well as some other methods for comparison.

Finally, it is worth reiterating here that the bottom-line performance difference between methods (e.g. between UMPIRE and its comparators) is *not* attributable to request frequencies alone—It also depends on *which* samples are being requested. Case in point: Purely based on request frequencies, some methods request less than UMPIRE, some methods request more than UMPIRE, and Matched Decaying Request requests just as frequently as UMPIRE. (Here, we also show an Epsilon-Request "x2" benchmark using twice the ($\epsilon$) rate of requests as the original Epsilon Request, as well as a Matched Decaying Request "x2" benchmark also using twice the ($\epsilon_t$) rate of requests as the original Matched Decaying Request). Yet we see in Figure 60(c)–(d) that UMPIRE still performs better than these comparators, whether they request at more, less, or the same rate. In fact, as we discuss in Section 4.2 and additionally shown in Appendix A.2, UMPIRE's advantage is due to a combination of cost-sensitivity, deliberate exploration, and uncertainty-awareness.

## E.8 Related Work: Human vs. AI to Defer

Literature related to this field can broadly be separated into (1) work that assumes a *machine* is in control, and selectively defers to an expert, and (2) work that assumes the *human* is in control, and selectively relies on machines for help.

The first perspective is often associated with the learning with rejection paradigm, as well as some active learning problem settings, such as those referenced and discussed in Section 2.2. The goal is often to optimize a certain performance metric of the model acting autonomously, with an important choice being when to defer to an expert.

The second perspective is often concerned with how humans themselves should best leverage machines to inform their own behavior in response. For instance, this is associated with "closing the loop" in clinical decision support [74], as well as how to select representative examples to teach the human how a machine learning model operates [102].

Indeed, while these two perspectives are often studied in separate clusters of work, they an and ought to be *complementary*—because in the real world, humans and machines must jointly act as an effective "team"/"system".

In our present work, the ODM objective is to minimize the cumulative *system regret* (in a stream-based setting with abstentive feedback), where an imperfect human is the initial decision-maker, and where a resource-scarce expert is available. In this sense, we are already moving from perspective (1) towards a more holistic view of humans and machines as a team (instead of just measuring model risk alone). In doing so, we focus on how a *mediator* should make suggestions, such that—if its recommendations were accepted—the system regret would be low.

In contrast, some other works have instead started from perspective (2), and then moved towards this more holistic view. In particular, it is important to account for the fact that the human indeed must *choose* whether to accept a model's recommendation, or make decisions on their own [103]—which has important implications for how to design model recommendations in the first place. Thus both perspectives are complementary, and future work would benefit from jointly studying how humans and machines should behave in a mutually-aware fashion [104].

## E.9 Performance w.r.t. Expert Stochasticity

Regarding the performance of UMPIRE in the presence of increasing expert stochasticity, note that the relative advantage between UMPIRE and *any* other benchmark that decreases as noise levels rise, as our estimates of uncertainty become more entangled with noise. This can be observed across the panels of Figures 22, 23, 24, and 25, where most methods begin to "bunch up" more tightly as we read the panels from left to right (beware of the vertical axis scaling). Since Pessimistic Bayesian Sampling generally performs quite well (on GaussSine), it is also true that the relative advantage between UMPIRE and Pessimistic Bayesian Sampling decreases as noise levels rise. However, this phenomenon is *not specific to* Pessimistic Bayesian Sampling. Instead, it is simply that the advantage conferred by UMPIRE's strategy is smaller as the inherent stochasticity of the expert increases.

This is relevant when considering how much stochasticity is present in different real-world settings. For instance, medical diagnosis in particular is known to be a fairly noisy process in the real world—especially in cases where an early diagnosis is required given limited information. In these cases, we would also expect that the advantage conferred by UMPIRE's strategy is smaller. In fact, this is what we can already observe from the performance results: We see slightly smaller advantages for UMPIRE in the Alzheimers and CysticFibrosis environments, relative to other benchmarks.

## E.10 Visualizing the Adjustment Factor

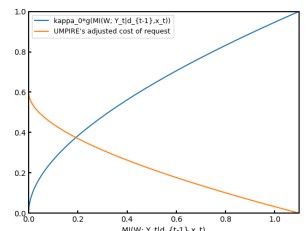

The right-hand-side of Equation 6 is a function of the mutual information ("MI") term $\mathbb{I}[W; Y_t | d_{t-1}, x_t]$, which represents how much the epistemic uncertainty in the model policy is expected to decrease if $Y_t \sim p(\cdot | d_{t-1}, x_t)$ is revealed. Here, we plot $\kappa_0 g(\mathbb{I}[W; Y_t | d_{t-1}, x_t])$ as a function of the inside MI term (using $m = 3$ as in the GaussSine environment). We confirm that it is monotonically increasing, which is in line with our motivation to "translate" the expected decrease in epistemic uncertainty into an upper bound on expected improvement in model risk. We also plot the resulting adjusted cost of request $\bar{k}_{\text{req}}$ (using the base value $k_{\text{req}} = 0.6$ as in the GaussSine environment). We see that this means $\bar{k}_{\text{req}}$ is low in the beginning when the MI term is high, whence UMPIRE behaves like standard incremental learning. In the limit of a perfect model, $\bar{k}_{\text{req}}$ converges to $k_{\text{req}}$ (towards the left side of the graph), whence UMPIRE behaves like the optimal greedy mediator.

## E.11 Applications of the ODM Framework

In addition to clinical decision support, the ODM problem setting is applicable to any scenario where "imperfect" decision-makers are the front-line decision-makers, and "oracle" decision-makers are available as expert supervision—albeit with limited availability, and where learning feedback is abstentive. This situation arises in many settings where the responsibility for a decision must ultimately fall on a *person* (i.e. the imperfect human or the expert), but a *machine* is available for learning and issuing recommendations. Some more examples, in addition to clinical decision support:

**Product Inspection**: A junior employee signs off on the quality of a product batch. The mediator can decide to (1) accept the sign-off *as is*, or (2) recommend a re-inspection due to a disagreeing autonomous prediction as to product quality, or (3) recommend that a more senior employee take over and issue their more qualified assessment.

**Content Moderation**: A user in a social network can report content violations in real time. The mediator can decide to (1) accept and act on the report, or (2) recommend that the user re-classify the content due to a disagreeing assessment as to its appropriateness, or (3) recommend that an internal moderator take over and issue their judgement.

**Spoken-Dialog System**: A customer selects a possibly-nonsensical option in a spoken-dialog system. The mediator can decide to (1) accept the user's option, or (2) recommend that the user re-select an option from the same menu, or (3) re-route the customer to a phone conversation with an actual (human) customer representative to continue the work.

Finally, note that ODM is also applicable in settings where there is no imperfect human involved, so the machine makes decisions with selective expert feedback. Consider setting $k_{\text{int}} = 0$ in ODM: This effectively recovers such a setting, where a machine interacts with an expert (i.e. the imperfect human becomes "redundant"). Note that this does not alter the "hardness" of the ODM problem, which is distinguished by the fact that expert feedback is costly and abstentive.

## E.12 Limitations of the ODM Framework

There are two main limitations of our analysis using the ODM framework:

First, ODM is an *online learning* framework. In general, it is known that online learning may not perform well during the early time steps when an online learner's decisions are largely exploratory, especially if learning proceeds "from scratch". In this sense, UMPIRE as a solution is also not immune to this challenge. Therefore, future work would benefit from examining the potential to *not* learn from scratch—e.g. to "warm-start" a learner using existing data, which can be done using a variety of methods from the extensive literature on imitation learning.

Second, we must recognize that there are *two sides* to human-machine interactions: While ODM focuses on how machines should best propose recommendations to humans, there is also the complementary aspect of how/whether humans actually incorporate such recommendations into their behavior. Ignoring this second aspect may lead to models that are accurate but not necessarily proposing recommendations that are most likely to be complied with—which would severely undermine the practical utility of such a model. Therefore, future work would also benefit from *jointly* studying how humans and machines should behave in a "mutually-aware" fashion.