# OpenReview forum: "Online Decision Mediation"
_NeurIPS.cc/2022/Conference — NeurIPS 2022 Accept_

### Official Review · Reviewer_X8r8 · 2022-06-16

**Rating:** 6
**Confidence:** 2
**Soundness:** 3 good
**Presentation:** 2 fair
**Contribution:** 3 good

**Summary:**

This paper studied a new problem: online decision mediation, in which the system (mediator policy) decides whether to follow human decision, or take an intervention, or request help from the expert based on human behaviors at every step. To learn a good mediator policy, the proposed solution UMPIRE balances the immediate loss terms against future improvement. They also validate the advantage of UMPIRE through empirical studies.

**Questions:**

Please refer to the weakness section. Moreover, Figures are too small to read.



**Limitations:**

Yes.

**Strengths And Weaknesses:**

Strength:
The big idea of online decision mediation framework is novel and interesting. Requesting advice from experts costs high in practice, while human behaviors are not always optimal and needs to be corrected if necessary. Thus, it’s very worthwhile to think about mediator policies. I think this paper proposed nice formulation and methodologies for this practical scenario.

Weakness:
1. Many parts are not clear.
    - in equation (3), what is Y?
    - How did you derive equation (5)?
2. The main theoretical result (Theorem 1) is hard to interpret. From the right side of equation (6), I cannot tell how much the improvement is. It will be helpful if authors can give some specific examples to show the scaling.
3. This paper is not easy to read and follow. Existing works and new techniques are often mixed in sections.

Overall though I think this paper is not that theoretically strong, but ODM and their methodologies are good contributions to the community.

---

> ### Author Response · Authors · 2022-07-31
> **Response to Reviewer X8r8**
>
> ---
>
> Thank you for your thoughtful comments and suggestions! Please find our answers as follows, along with corresponding updates to the revised submission:
>
> ---
>
> **(A) Equation 3 Clarification**
>
> We agree the meaning of "$Y$" here can be clarified.
>
> To be clear, $\delta(Y-\tilde{Y}_t)$ is simply the Dirac delta centered at $\tilde{Y}_t$, so "$Y$" is just the free variable of this distribution. This is the same as its usage in Equation 1 (explained on line 83).
>
> **UPDATE**: There was a typo above Equation 3, which may have contributed to the confusion: In line 99, the terms "$X,Y$" should instead read "$\mathcal{X},\mathcal{Y}$". This is now fixed.
>
> ---
>
> **(B) Equation 5 Clarification**
>
> We agree this equation can be explained more intuitively.
>
> By definition, the greedy policy $\phi_{*}$ is simply one that minimizes the one-step system risk. In other words, it compares the following quantities:
>
> - $1-\hat{\pi}(\tilde{Y}|X)$
>
> - $1-\hat{\pi}(\hat{Y}|X)+k_{\text{int}}$
>
> - $k_{\text{req}}$
>
> and selects $Z=0$ if the first is smallest, $Z=1$ if the second is smallest, and $Z=2$ if the third is smallest (with ties---if they exist---broken arbitrarily). This follows immediately from the definition of system risk (Definition 2), and Equation 5 is simply a more concise way to express this policy.
>
> **UPDATE**: In the camera-ready version, this exact explanation above will be included in the text immediately following Equation 5. Due to the 9-page limit for draft submissions, we will defer this update to the camera-ready version, since it requires a small paragraph's worth of space. However, in the camera-ready version the extra 10-th page allowed means this will fit comfortably in the main manuscript.
>
> ---
>
> **(C) Equation 6 Visualization**
>
> Thank you for the suggestion---Indeed, visualizing the right-hand-side of Equation 6 is beneficial.
>
> Briefly, the right-hand-side is a function of the mutual information ("MI") term $\mathbb{I}[W;Y_{t}|d_{t-1},x_{t}]$. This term itself should be intuitive: As discussed in the manuscript, it represents how much the epistemic uncertainty in the model policy is expected to decrease if $Y_{t}\sim p(\cdot|d_{t-1},x_{t})$ is revealed. But since this function involves an exponential and product logarithm function, we agree that it may not be immediately clear how it looks. We have therefore now provided a concrete visualization:
>
> **UPDATE**: In a (new) **Appendix E.10** ("Visualizing the Adjustment Factor"), we now included a figure that plots $\kappa_{0}g(\mathbb{I}[W;Y_{t}|d_{t-1},x_{t}])$ as a function of the inside mutual information term (using $m=3$ as in the GaussSine environment). We confirm that it is monotonically increasing, which is in line with our motivation to "translate" the expected decrease in epistemic uncertainty into an upper bound on expected improvement in model risk. Perhaps more importantly, we also plot the resulting adjusted cost of request $\\bar{k}\_{\\text{req}}$ (using the base value $k_{\\text{req}}=0.6$ as in the GaussSine environment). We see that this means $\\bar{k}\_{\\text{req}}$ is low in the beginning when the mutual information term is high, whence UMPIRE behaves like standard incremental learning. In the limit of a perfect model, $\\bar{k}\_{\\text{req}}$ converges to $k\_{\\text{req}}$, whence UMPIRE behaves like the optimal greedy mediator.
>
> ---
>
> **(D) Readability**
>
> We agree that the figures are currently quite small.
>
> **UPDATE**: In the camera-ready version, we will make the figures larger using the extra 10-th page allowed.
>
> Finally, note that the only places we discuss existing works alongside new techniques is in Section 2.2 (related work, which is where we distinguish the ODM problem from related but conceptually distinct settings), and in Section 3.1 (bandit policies, which is where we highlight why bandit policies may appear to be an obvious solution, but actually suffer in ODM). In all other parts of the manuscript, there should be no mixing of new/existing work.

---

### Official Review · Reviewer_Vsuz · 2022-07-11

**Rating:** 5
**Confidence:** 4
**Soundness:** 3 good
**Presentation:** 4 excellent
**Contribution:** 2 fair

**Summary:**

The work proposes a solution to the following problem setting: Given an input instance, an imperfect human provides a provisional decision/prediction. A mediator model decides whether to accept, to intervene by showing its own decision (at a cost), or whether to defer (at a cost) to another human expert. The instances come in a streaming fashion. The proposed approach (UMPIRE) trades off system risk (final decision) for model risk (ability of the mediator to make predictions and learn about the deferring strategy). The algorithm is compared with several related works, mainly bandit algorithms, and shown to mostly outperform. A detailed sensitivity analysis is also presented.

**Questions:**

My questions are mostly related to W1 above. The human appears to be somewhat redundant in the workflow, also because they seem to have no say if their decision is overridden by the model's decision. What is the motivation for choosing \alpha = 0.5?

In addition, it would be useful to see the progress of how z is distributed across timesteps. Does the model decide to request for the expert's prediction mostly at the beginning? When does it start to request less and in which cases in the timeline it decides to just accept the human decision?

**Limitations:**

-Literature in this field is somewhat divided between work that assumes that a model does the routing (deferring) and work that assumes that the human is in control for when to defer to an AI or when to ask for help or when to decide to integrate AI recommendations in their decision (see examples below).  There probably needs to be a discussion here or in the introduction that describes when\why it makes sense to sometimes have a model that controls the routing.

Teaching Humans When To Defer to a Classifier via Exemplars

Is the most accurate ai the best teammate? optimizing ai for teamwork

Updates in human-ai teams: Understanding and addressing the performance/compatibility tradeoff

- There may also need to be a more extended discussion on why pessimistic bayesian sampling becomes a competitive option when the expert noise increases. This detail also needs to be mentioned in the introduction or abstract as an important insight for real-world applications, where indeed most often than not expert decisions are also not perfect.

**Strengths And Weaknesses:**

Strengths:

S1- Optimizing for overall system accuracy instead of autonomous model predictions is an important albeit overlooked direction.

S2- The paper is very well written, and results are clearly presented.

S3- The paper also does a very good job at contrasting and comparing with related work.

Weaknesses:
W1- While optimizing for overall system accuracy is important, the choice of keeping an imperfect human decision maker in the loop is not very well motivated from both an ethical and mathematical standpoint. If I am not mistaken, the imperfect human decision maker is simulated with a probability of 0.5 (line 304). Thus, it is questionable why keep this actor in the loop on the first place and why not change the whole interaction altogether to have the model interact only with the expert. This is in fact the setting that other works in human-ai decision making also consider (e.g. [24] non-streaming or [*] streaming). Otherwise, if there are any other reasons to why this workflow may still make sense, at least the experiments need to be adjusted to a probability of error > 0.5 for the human.

[24] Consistent estimators for learning to defer to an expert
[*] Pre-emptive learning-to-defer for sequential medical decision-making under uncertainty

---

> ### Author Response · Authors · 2022-07-31
> **Response to Reviewer Vsuz [Part 4/4]**
>
> ---
>
> ### **(D) Performance w.r.t. Expert Stochasticity**
>
> We assume you are referring to Figure 5 ("Expert Stochasticity vs. Performance"), or---in more detail---Figures 22--25.
>
> Kindly allow us to point out that it is the relative advantage between UMPIRE and **any** other benchmark that decreases as noise levels rise. This can be observed across the panels of Figures 22--25, where most methods begin to "bunch up" more tightly as we read the panels from left to right (beware of the vertical axis scaling). Since Pessimistic Bayesian Sampling generally performs quite well (on GaussSine), it is also true that the relative advantage between UMPIRE and Pessimistic Bayesian Sampling decreases as noise levels rise. However, this phenomenon is **not specific to** Pessimistic Bayesian Sampling. Instead, it is simply that the advantage conferred by UMPIRE's strategy is smaller as the inherent stochasticity of the expert increases.
>
> In any case, we agree that a mention of stochasticity in the **real world** is beneficial. For instance, medical diagnosis in particular is known to be a fairly noisy process in the real world---especially in cases where an early diagnosis is required given limited information. This may also be true in online content moderation, product inspection, etc. as well. In these cases, we would also expect that the advantage conferred by UMPIRE's strategy is smaller, as our estimates of uncertainty become more entangled with noise. In fact, this is what we can already observe from the performance results: We see slightly smaller advantages for UMPIRE in the Alzheimers and CysticFibrosis environments, relative to other benchmarks.
>
> **UPDATE**: In the camera-ready paper, a version of the above discussion can be readily integrated into Section 4.3 ("Sensitivity Analysis") in the main manuscript using the 10-th camera-ready page. Due to the 9-page limit for drafts, we have currently included the above discussion in an **Appendix E.9** ("Performance w.r.t. Expert Stochasticity").
>
> ---
>
> ### **References**
>
> Throughout the new discussion sections included in the revised appendices, we have now included the following suggested references to our work (numbering continued from bibliography in appendices):
>
> - [101] Joshi et al. Pre-emptive Learning-to-Defer for Sequential Medical Decision-Making under Uncertainty.
> - [102] Mozannar et al. (AAAI 2022). Teaching Humans When to Defer to a Classifier via Exemplars.
> - [103] Bansal et al. (AAAI 2021). Is the Most Accurate AI the Best Teammate? Optimizing AI for Teamwork.
> - [104] Bansal et al. (AAAI 2019). Updates in Human-AI Teams: Understanding and Addressing the Performance/Compatibility Tradeoff.

---

> ### Author Response · Authors · 2022-07-31
> **Response to Reviewer Vsuz [Part 3/4]**
>
> ---
>
> ### **(B) How Mediation Evolves over Time**
>
> We agree that it is beneficial to visualize how mediation actions $Z$ evolve over time. You are correct---Intuitively, we would expect a good mediator policy to request more in the beginning, when the model is more likely to learn from expert actions (and more likely to make mistakes otherwise).
>
> **UPDATE**: First, we have now included a plot of how $Z$ evolves over time, in Figure 60(a) in a (new) **Appendix E.7** ("How Mediation Evolves over Time"). This visualizes the relative frequencies of each mediator action (i.e. "accept", "intervene", and "request") that UMPIRE issues, using GaussSine as environment (plots are similar for other environments). For less noisy visualization, numbers are plotted as moving averages of windows of length 10 (steps). Second, we can also ask how the pattern of "request" actions compares across different methods. Figure 60(b) plots the cumulative progression of request actions over time for UMPIRE, as well as some other methods for comparison.
>
> Finally, it is worth reiterating here that the bottom-line performance difference between methods (e.g. between UMPIRE and its comparators) is **not** attributable to request frequencies alone---It also depends on **which** samples are being requested. Case in point: Purely based on request frequencies, some methods request less than UMPIRE, some methods request more than UMPIRE, and Matched Decaying Request requests just as frequently as UMPIRE. (Here, we also show an Epsilon-Request "x2" benchmark using twice the ($\epsilon$) rate of requests as the original Epsilon Request, as well as a Matched Decaying Request "x2" benchmark also using twice the ($\epsilon_{t}$) rate of requests as the original Matched Decaying Request). Yet we see in Figure 60(c)--(d) that UMPIRE still performs better than these comparators, whether they request at more, less, or the same rate. In fact, as we discuss in Section 4.2 and additionally shown in Appendix A.2, UMPIRE's advantage is due to a combination of cost-sensitivity, deliberate exploration, and uncertainty-awareness.
>
> ---
>
> ### **(C) Related Work: Human vs. AI to Defer**
>
> We agree with your analysis that literature related to this field can broadly be separated into (1) work that assumes a **machine** is in control, and selectively defers to an expert, and (2) work that assumes the **human** is in control, and selectively relies on machines for help:
>
> - The first perspective is often associated with the *learning with rejection* paradigm, as well as some *active learning* problem settings, such as those referenced and discussed in Section 2.2. The goal is often to optimize a certain performance metric of the model acting autonomously, with an important choice being when to defer to an expert.
>
> - The second perspective is often concerned with how humans themselves should best leverage machines to inform their own behavior in response. For instance, this is associated with *closing the loop* in clinical decision support [74], as well as how to select representative examples to *teach the human* how a machine learning model operates [102].
>
> Indeed, while these two perspectives are often studied in separate clusters of work, they an and ought to be **complementary**---because in the real world, humans and machines must jointly act as an effective "team"/"system".
>
> - In our present work, the ODM objective is to minimize the cumulative *system regret* (in a stream-based setting with abstentive feedback), where an imperfect human is the initial decision-maker, and where a resource-scarce expert is available. In this sense, we are already moving from perspective (1) towards a more holistic view of humans and machines as a team (instead of just measuring model risk alone). In doing so, we focus on how a *mediator* should make suggestions, such that---if its recommendations were accepted---the system regret would be low.
>
> - As you kindly suggest, some other works have instead started from perspective (2), and then moved towards this more holistic view. In particular, it is important to account for the fact that the human indeed must *choose* whether to accept a model's recommendation, or make decisions on their own [103]---which has important implications for how to design model recommendations in the first place. Thus both perspectives are complementary, and future work would benefit from jointly studying how humans and machines should behave in a mutually-aware fashion.
>
> **UPDATE**: In the camera-ready paper, a version of the above discussion can be readily integrated into Section 5 ("Conclusion") in the main manuscript using the 10-th camera-ready page. Due to the 9-page limit for drafts, we have currently included the above discussion in an **Appendix E.8** ("Related Work: Humans vs. AI to Defer").

---

> ### Author Response · Authors · 2022-07-31
> **Response to Reviewer Vsuz [Part 2/4]**
>
> ---
>
> **(A.3) Sensitivity on probability of human error**
>
> Yes, in our experiments the imperfect human's decisions are simulated with $\alpha=0.5$, although the specific frequency of human mistakes does not affect the basic structure/hardness of the problem. Our motivation is simply to simulate decisions that stochastically deviate from the expert's with some probability between zero and one.
>
> We agree that it is beneficial to verify that our intuitions still hold (and that UMPIRE still performs as expected) under a higher probability of error for the human. First, as you suggest, we performed a complete re-run of our main experiments under the same conditions as before---but now setting $\alpha=0.9$.
>
> **UPDATE**: We have now included this new set of results in a (new) **Appendix E.3** ("Sensitivity on Human Error (Performance)") and (new) **Appendix E.4** ("Sensitivity on Human Error (Source of Gain)"). This includes the following results, each reported separately for the (a) GaussSine, (b) HighEnergy, (c) MotionCapture, (d) LunarLander, (e) Alzheimers, and (f) CysticFibrosis environments [F2]:
>
> (respectively for performance and source-of-gain analysis):
>
> - Figures 32 and 39: System Performance (Losses)
> - Figures 33 and 40: System Performance (Regrets)
> - Figures 34 and 41: System Performance (Mistakes)
> - Figures 35 and 42: Model Performance (Heldout Mistakes)
> - Figures 36 and 43: Model Performance (Heldout CrossEnt)
> - Figures 37 and 44: Model Performance (Heldout AUROC)
> - Figures 38 and 45: Model Performance (Heldout AUPRC)
>
> **UPDATE**: For completeness, we can additionally ask how performance varies for different levels of $\alpha$. Moreover, in light of the discussion in Response (A.2) above, we can also ask what performance looks like if $k_{\text{int}}=0.0$ instead of $k_{\text{int}}=0.1$. To this end, we have now also included complete sensitivities for the cartesian product of settings $\alpha\in\{0.5,0.7,0.9,1.0\}$ and $k_{\text{int}}\in\{0.0,0.1\}$. For this, we focus on the GaussSine environment to keep experiments tractable within the response period. The results are now included in a (new) **Appendix E.5** ("Complete Sensitivities on $\alpha$ and $k_{\text{int}}$ (Performance)") and (new) **Appendix E.6** ("Complete Sensitivities on $\alpha$ and $k_{\text{int}}$ (Source of Gain)"). As above, these include:
>
> (respectively for performance and source-of-gain analysis):
>
> - Figures 46 and 53: System Performance (Losses)
> - Figures 47 and 54: System Performance (Regrets)
> - Figures 48 and 55: System Performance (Mistakes)
> - Figures 49 and 56: Model Performance (Heldout Mistakes)
> - Figures 50 and 57: Model Performance (Heldout CrossEnt)
> - Figures 51 and 58: Model Performance (Heldout AUROC)
> - Figures 52 and 59: Model Performance (Heldout AUPRC)
>
> Across all sensitivities, it is easy to verify that UMPIRE still consistently accumulates lower regret (our primary metric of interest), as well as outperforming comparators with respect to to the rest of the performance measures.
>
> [F2] We do not include Matched Decaying Request in these additional experiments, because each run requires a post-hoc search---for each and every setting---for a polynomial function that best models UMPIRE's request pattern, which is intractable within the response period. In any case, we have already empirically demonstrated that, indeed, even (artificially and unfairly) matching UMPIRE's request frequencies is insufficient to perform as well as UMPIRE (see Section 4.2 and Appendix A.2).

---

> ### Author Response · Authors · 2022-07-31
> **Response to Reviewer Vsuz [Part 1/4]**
>
> ---
>
> Thank you for your thoughtful comments and suggestions! Please find our answers as follows, along with corresponding updates to the revised submission:
>
> - (A) Keeping the Imperfect Human in the Loop
> - (B) How Mediation Evolves over Time
> - (C) Related Work: Human vs. AI to Defer
> - (D) Performance w.r.t. Expert Stochasticity
>
> ---
>
> ### **(A) Keeping the Imperfect Human in the Loop**
>
> ---
>
> **(A.1) Why keep the imperfect human in the loop?**
>
> Actually, our choice to keep the imperfect human in the loop is indeed motivated by real-world ethical considerations: In many high-stakes settings, concrete attribution of **responsibility** is an absolute requirement. For instance, in most healthcare systems, the responsibility for a cancer diagnosis must be traceable to the **person** (not a **machine**) who actually ordered it---and is therefore legally/ethically/financially accountable for it (see e.g. [13] and [15]).
>
> Our formulation of the Online Decision Mediation ("ODM") problem reflects this consideration. In our framework, an imperfect human first attempts to make a decision, which is observed. Then, the **mediator** decides which of the following happens:
>
> - "Accept": The imperfect human's decision goes through. So the **imperfect human** is responsible for that decision.
>
> - "Intervene": The mediator proposes an alternative to the imperfect human. This immediately incurs a cost ($k_{\text{int}}$), representing the fact that the imperfect human is now asked to spend time/resources in reconsidering/altering their original decision in light of the proposal. But the **imperfect human** is still responsible for the decision, regardless of whether or not they comply with the proposal [F1].
>
> - "Request": The expert is invited to make the decision instead. So the **expert** is now responsible for that decision.
>
> [F1] Whether or not they comply with the proposal is beyond the scope of our work: There may be a variety of reasons why they do/don't comply downstream. Importantly, however, from the perspective of the **mediator**, it has already done its job---and the correctness of its proposed alternative can be evaluated (and on the basis of which we can define concrete notions of model regret and system regret, which is what we do).
>
> If we "remove" the imperfect human from the loop, then all decisions would be made autonomously by the machine unless the expert is queried---which is strongly at odds with societal notions of ethical responsibility/accountability for high-stakes decisions. (To be clear, the "intervene" option is not autonomously "overriding" the imperfect human's decision: It is simply proposing an alternative!)
>
> **UPDATE**: We agree that this motivation is important to clarify, and we agree that it may not be apparent from the mostly mathematical description that is Section 2.1 ("Problem Formulation"). We have now included a version of this discussion in a (new) **Appendix E.1** ("Why keep the imperfect human in the loop?").
>
> ---
>
> **(A.2) Can the model interact only with the expert?**
>
> You are correct---Many existing works operate in a framework where the **machine** itself is directly allowed to make decisions autonomously, and selectively query the expert (e.g. when it is uncertain). Given the above discussion, we would expect that this setting is generally applicable in non-high-stakes decision problems (i.e. where notions of responsibility are not legally/ethically/financially required to be tied to a **person**).
>
> That being said, the ODM framework is simply a **generalization** of this. Consider setting $k_{\text{int}}=0$ in ODM: This effectively recovers the simpler setting where a machine interacts with an expert (i.e. the imperfect human would indeed become "redundant"). Importantly, however, in general the human is *not* "redundant" whenever $k_{\text{int}}>0$, as long as the imperfect human has non-zero probability of being correct. Precisely, the mediator policy needs to decide if the human is likely already correct, so it can choose to "accept" (instead of "intervene") in order to avoid incurring the cost $k_{\text{int}}$.
>
> Finally, it is worth reiterating that the ODM problem is also distinguished by the fact that feedback is **abstentive**. Whereas, existing works in machine-expert interaction (including [24] and [101] that you kindly referenced) operate in settings where feedback is **not** abstentive. That is to say, even if we set $k_{\text{int}}=0$, the ODM setting is still fundamentally more challenging to tackle, as discussed throughout the manuscript.
>
> **UPDATE**: We agree that this aspect of relationship with existing work is important to clarify. We have now included a version of this discussion in a (new) **Appendix E.2** ("Can the model interact only with the expert?").

---

> ### Author Response · Authors · 2022-08-08
> **Dear Reviewer Vsuz**
>
> ---
>
> Thank you again for your time and expertise during the review process!
>
> If there were any leftover concerns, we would sincerely appreciate the opportunity to clarify them---before the discussion period for authors ends. We believe our responses (Aug 1) have addressed in detail the full set of questions you had raised:
>
> - Keeping the imperfect human in the loop; see **Response (A)**
> - How mediation actions evolve over time; see **Response (B)**
> - Related work on human vs. AI to defer; see **Response (C)**
> - Performance wrt. expert stochasticity; see **Response (D)**
>
> These are paired with corresponding updates to the submission (Aug 1), including extensive additional experiment results, sensitivity analyses, and discussions and clarifications:
>
> - (new) **Appendix E.1** ("Why keep the imperfect human in the loop?")
> - (new) **Appendix E.2** ("Can the model interact only with the expert?")
> - (new) **Appendix E.3** ("Sensitivity on Human Error (Performance)")
> - (new) **Appendix E.4** ("Sensitivity on Human Error (Source of Gain)")
> - (new) **Appendix E.5** ("Complete Sensitivities on $\alpha$ and $k_{\text{int}}$ (Performance)")
> - (new) **Appendix E.6** ("Complete Sensitivities on $\alpha$ and $k_{\text{int}}$ (Source of Gain)")
> - (new) **Appendix E.7** ("How Mediation Evolves over Time")
> - (new) **Appendix E.8** ("Related Work: Humans vs. AI to Defer")
> - (new) **Appendix E.9** ("Performance w.r.t. Expert Stochasticity")
> - (new) **References** (all suggested references are now included)
>
> We would appreciate if the reviewer kindly let us know if there were any further questions in the very limited time remaining. We are eager to do our utmost to address them!
>
> Thank you,
>
> Paper8503 Authors

---

### Official Review · Reviewer_EaPK · 2022-07-12

**Rating:** 7
**Confidence:** 3
**Soundness:** 3 good
**Presentation:** 3 good
**Contribution:** 3 good

**Summary:**

An interesting paper which focuses on formalising the sequential problem of online decision mediation (ODM)  - and proposes a specialised solution (uncertainty-modulated policy for intervention and requisition - UMPIRE) to tackling pitfalls which traditional bandit algorithms face in ODM. The paper is relevant and very promising for describing potential solutions for real-world and real-time decision making, especially in safety-critical applications (e.g. clinical diagnosis), wherein, a human-in-the-loop approach can benefit from providing an effective interface between  human mistakes and expert feedback - as the paper also mentions.

**Questions:**

Where could this proposed solution be used outside of clinical decision support? Can the authors provide few more examples in the Conclusions section e.g. could it be used in industry for real-time decision support of engineering systems (e.g. during operations & maintenance) etc.?


**Limitations:**

What are the main drawbacks/limitations of UMPIRE? It is not clearly mentioned in the paper's Section 5 - it would have been useful to have a separate "sub-heading" for Limitations e.g. in a Discussion section as it is inadvertent that the proposed algorithm may suffer from pitfalls which need to be clearly outlined with simplistic language (and comprehensively).

Update: Thanks for addressing this reviewer's comments - the paper is now substantially clearer and comprehensive. I have changed my score from 6 to 7.

**Strengths And Weaknesses:**

A very well written and comprehensive paper, well-supported by experiments and comparison with other baseline models/algorithms, particularly on real-world datasets/benchmarks which makes it very interesting to read.

The main weakness of the paper is that it is very verbose and text-heavy at some places e.g. in Section 2.2. Can these e.g. "learning with rejection" be better encapsulated (or relevant papers from past literature cited instead of a complete re-definition from scratch?). This would greatly help improve readability of the paper.

---

> ### Author Response · Authors · 2022-07-31
> **Response to Reviewer EaPK [Part 2/2]**
>
> ---
>
> **(B) Main Limitations**
>
> We agree that a separate "Limitations" sub-heading is beneficial. There are two main limitations of our analysis using the ODM framework:
>
> - First, ODM is an **online learning** framework. In general, it is known that online learning may not perform well during the early time steps when an online learner's decisions are largely exploratory, especially if learning proceeds "from scratch". In this sense, UMPIRE as a solution is also not immune to this challenge. Therefore, future work would benefit from examining the potential to **not** learn from scratch---e.g. to "warm-start" a learner using existing data, which can be done using a variety of methods from the extensive literature on imitation learning.
>
> - Second, we must recognize that there are **two sides** to human-machine interactions: While ODM focuses on how machines should best propose recommendations to humans, there is also the complementary aspect of how/whether humans actually incorporate such recommendations into their behavior. Ignoring this second aspect may lead to models that are accurate but not necessarily proposing recommendations that are most likely to be complied with---which would severely undermine the practical utility of such a model. Therefore, future work would also benefit from **jointly** studying how humans and machines should behave in a "mutually-aware" fashion.
>
> **UPDATE**: In the camera-ready paper, a version of the above discussion can be readily integrated into Section 5 ("Conclusion") in the main manuscript using the 10-th camera-ready page. Due to the 9-page limit for drafts, we have currently included the above discussion in an **Appendix E.12** ("Limitations of the ODM Framework").
>
> ---
>
> **(C) Readability**
>
> We agree that Section 2.2 (which primarily serves as our related works section) is rather text-heavy. However, there is indeed a **wide** variety of related topics that could bear some (superficial) resemblance to the ODM formalism. So, in order to clearly distinguish these related fields from our novel setting, it is inevitable that a lot of ground must be covered at once.
>
> **UPDATE**: To make it clear that Section 2.2 primarily serves as the related works section, we have now changed the Section 2.2 title to be "Related Work". This should make it clear that readers have the option of referring to it in a non-chronological fashion---so they can choose to first read the rest of the paper before coming back for the detailed comparison with related works.

---

> > ### Comment · Reviewer_EaPK · 2022-08-08
> > **Thanks for your response**
> >
> > Update: Thanks for addressing this reviewer's comments - the paper is now substantially clearer and comprehensive. I have changed my score from 6 to 7.

---

> ### Author Response · Authors · 2022-07-31
> **Response to Reviewer EaPK [Part 1/2]**
>
> ---
>
> Thank you for your thoughtful comments and suggestions! Please find our answers as follows, along with corresponding updates to the revised submission:
>
> - (A) Other Applications
> - (B) Main Limitations
> - (C) Readability
>
> ---
>
> **(A) Other Applications**
>
> We agree that giving additional examples of possible applications is beneficial.
>
> The ODM problem setting is applicable to any scenario where "imperfect" decision-makers are the front-line decision-makers, and "oracle" decision-makers are available as expert supervision---albeit with limited availability, and where learning feedback is abstentive. This situation arises in many settings where the responsibility for a decision must ultimately fall on a *person* (i.e. the imperfect human or the expert), but a *machine* is available for learning and issuing recommendations. Here are some more examples, in addition to clinical decision support:
>
> - **Product Inspection**: A junior employee signs off on the quality of a product batch. The mediator can decide to (1) accept the sign-off *as is*, or (2) recommend a re-inspection due to a disagreeing autonomous prediction as to product quality, or (3) recommend that a more senior employee take over and issue their more qualified assessment.
>
> - **Content Moderation**: A user in a social network can report content violations in real time. The mediator can decide to (1) accept and act on the report, or (2) recommend that the user re-classify the content due to a disagreeing assessment as to its appropriateness, or (3) recommend that an internal moderator take over and issue their judgement.
>
> - **Spoken-Dialog System**: A customer selects a possibly-nonsensical option in a spoken-dialog system. The mediator can decide to (1) accept the user's option, or (2) recommend that the user re-select an option from the same menu, or (3) re-route the customer to a phone conversation with an actual (human) customer representative to continue the work.
>
> We also agree that decision support of engineering systems during operations/maintenance would be another applicable example---for instance, as a variation/adaptation of the first example above.
>
> Finally, note that ODM is also applicable in settings where there is no imperfect human involved, so the machine makes decisions with selective expert feedback. Consider setting $k_{\text{int}}=0$ in ODM: This effectively recovers such a setting, where a machine interacts with an expert (i.e. the imperfect human becomes "redundant"). Note that this does not alter the "hardness" of the ODM problem, which is distinguished by the fact that expert feedback is costly and abstentive.
>
> **UPDATE**: In the camera-ready paper, a version of the above discussion can be readily integrated into Section 5 ("Conclusion") in the main manuscript using the 10-th camera-ready page. Due to the 9-page limit for drafts, we have currently included the above discussion in an **Appendix E.11** ("Applications of the ODM Framework").

---

### Meta-Review · Area_Chair_sejF · 2022-08-26

**Recommendation:** Accept
**Confidence:** Certain

**Metareview:**

Thank you for submitting your paper to NeurIPS! This paper studies sequential decision-making for mediation -- given actions chosen by an imperfect human, it decides whether to accept the decision, intervene with an alternative, or request a costly expert opinion. The authors identify an exploration-exploitation tradeoff for this problem (costly to obtain labels, but improves future accuracy) and build on the bandit literature to propose the new UMPIRE policy. The model and algorithmic approach seem sound for optimizing overall system accuracy, and the reviewers especially appreciated the extensive experiments on real datasets. I am pleased to recommend acceptance. However, please be sure to add more discussion of the limitations into the main paper (as promised in the response); even if the entire discussion doesn't fit in the main text, please add pointers in the main paper to Appendix E.

There are also some technical choices that were brought up by the senior program committee that would be useful to discuss since they may have usability/impact implications. First, what are implications of the choice of a scalarized objective that trades off the error metric with the expert cost (Eq. 3), say compared with something that's based on constraints? Does the scalarized objective do a good job approximating how we expect decisions to be triaged in practice? In what applications? Second, the regime k_req =m/m-1 - gamma is mentioned as interesting (in Remark 1) when combined with a 0/1 loss. Could you please expand on applications where the 0/1 loss is appropriate and the interpretation of the “interesting regime” in that case?

**Award:**

No

---

### Decision · Program_Chairs · 2022-09-14

Accept